# Intrinsic TGF-β signaling attenuates proximal tubule mitochondrial injury and inflammation in chronic kidney disease

Merve Kayhan [1], Judith Vouillamoz [1], Daymé Gonzalez Rodriguez[2], Milica Bugarski[3], Yasutaka Mitamura[4], Julia Gschwend [1], Christoph Schneider [1], Andrew Hall [3], David Legouis [5], Cezmi A. Akdis[4], Leary Peter [2], Hubert Rehrauer [2], Leslie Gewin [6,7], Roland H. Wenger [1] & Stellor Nlandu Khodo [1] ✉

Excessive TGF-β signaling and mitochondrial dysfunction fuel chronic kidney disease (CKD) progression. However, inhibiting TGF-β failed to impede CKD in humans. The proximal tubule (PT), the most vulnerable renal segment, is packed with giant mitochondria and injured PT is pivotal in CKD progression. How TGF-β signaling affects PT mitochondria in CKD remained unknown. Here, we combine spatial transcriptomics and bulk RNAseq with biochemical analyses to depict the role of TGF-β signaling on PT mitochondrial home-ostasis and tubulo-interstitial interactions in CKD. Male mice carrying specific deletion of *Tgfbr2* in the PT have increased mitochondrial injury and exacer-bated Th1 immune response in the aristolochic acid model of CKD, partly, through impaired complex I expression and mitochondrial quality control associated with a metabolic rewiring toward aerobic glycolysis in the PT cells. Injured S3T2 PT cells are identified as the main mediators of the maladaptive macrophage/dendritic cell activation in the absence of *Tgfbr2*. snRNAseq database analyses confirm decreased TGF-β receptors and a metabolic deregulation in the PT of CKD patients. This study describes the role of TGF-β signaling in PT mitochondrial homeostasis and inflammation in CKD, sug-gesting potential therapeutic targets that might be used to mitigate CKD progression.

Renal metabolic reprogramming and inflammation accompany the progression of chronic kidney disease (CKD)[1,2], which affects about 10% of humans worldwide and increases the risk of cardiovascular disease[3–5]. Excessive TGF-β signaling and mitochondrial dysfunction enhance the progression of CKD[6–11].

Transforming growth factor beta (TGF-β) is a pleiotropic factor that promotes renal fibrosis in CKD. Epithelial TGF-β signaling requires the binding activation of a serine/threonine kinase receptor, the TGF-β type II receptor (Tgfbr2 or TβRII) which subsequently activates the type I receptor (ALK5) and downstream SMAD-dependent and

[1]Institute of Physiology, University of Zurich, Zurich, Switzerland. [2]Functional Genomics Center Zurich, University of Zurich and ETH Zurich, Zurich, Switzerland. [3]Institute of Anatomy, University of Zurich, Zurich, Switzerland. [4]Swiss Institute of Allergy and Asthma Research, University of Zurich, Zurich, Switzerland. [5]Laboratory of Nephrology, Department of Medicine and Cell Physiology, Hospital and University of Geneva, Geneva, Switzerland. [6]Department of Internal Medicine, Division of Nephrology, Washington University, St. Louis, USA. [7]Department of Medicine, St. Louis Veterans Affairs, St. Louis, USA. ✉e-mail: stellor.nlandukhodo@uzh.ch

-independent effectors leading to the expression of diverse TGF-β target genes[12–14]. The TGF-β type III receptor is a membrane proteoglycan that acts as a co-receptor with other TGF-β receptors[15]. TGF-β signaling regulates a broad spectrum of biological processes involved in tissue homeostasis and injury response including cell growth and differentiation, migration, survival, and death[14,16–19]. TGF-β is a multi-faceted cytokine that mediates pro-and anti-inflammatory responses depending on the microenvironment and the targeted cell types. However, global TβRII or TGF-β1 knockout cause lethal inflammatory disorders in mice[20–23]. Despite its notoriety in promoting renal fibrosis, the hallmark of CKD progression, pharmacological inhibition of TGF-β signaling has not yet translated into successful therapy in humans[19]. We previously demonstrated that genetic deletion of TβRII in the proximal tubule (PT), the most metabolic and vulnerable renal segment, aggravates cortical fibrosis in two models of CKD[24]. However, cellular mechanisms underlying TGF-β's beneficial effect remained unclear.

Mitochondrial dysfunction has been implicated in a broad range of inherited and acquired renal diseases, including tubular defects (Fanconi and Bartter-like syndromes), cystic disease, acute kidney injury, and glomerular diseases[25]. Numerous studies have reported disruption of mitochondrial respiration in CKD, notably inactivation of complex IV in CKD patients[7,26]. Mitochondria dysfunction in podocytes promotes glomerular diseases and proteinuria[27,28], and uremic toxins reportedly impair electron transport chain (ETC) function and cause cell dedifferentiation[29].

The PTs constitute the most sensitive renal segments to injury partly due to its high metabolic rate and oxygen dependency, but also its higher exposure to diverse toxins. PT cells rely on their abundant mitochondria and oxidative phosphorylation (OXPHOS) to generate the energy needed to support their re-absorptive function. To survive and preserve their function, PT cells require a sophisticated mitochondrial quality control including biogenesis, mitophagy, and reactive oxygen species (ROS) buffering system. Injured PT cells acquire a secretory phenotype and play a pivotal role in the pathogenesis of CKD, likely through the paracrine effects of their secretome on the renal interstitium[30,31].

To define how TGF-β signaling affects PT mitochondrial homeostasis and tubulo-interstitial interactions in CKD, we deleted TβRII in the PT, injured mice with aristolochic acid (AA), a nephrotoxic compound causing PT injury and CKD in humans, and investigated mitochondrial integrity and inflammation. This study demonstrates that deleting TβRII in the PT worsens mitochondrial injury, partly by disrupting mitochondrial quality control, leads to a metabolic switch toward aerobic glycolysis (Warburg-like effect) and exacerbates Th1 inflammatory response in CKD.

## Results

### Proximal tubule TβRII deletion enhances renal remodeling and susceptibility to CKD

Injured PT cells play a pivotal role in the pathogenesis of CKD. To test how PT TGF-β signaling affects tissue remodeling and CKD progression, we injured mice with AA and analyzed renal injury and fibrosis using spatial transcriptomics (Visium) and biochemical approaches. Visium dataset integration and cluster annotation using the GSE151658 database[32] identified the main renal cell types (Fig. 1a and Supplementary Fig. 1a–d). Cluster resolution on kidney slices revealed impaired renal cortico-medullary organization in injured kidneys as compared to uninjured kidneys (Fig. 1b, Supplementary Figs. 1 and 2 and Supplementary Table 3). Consistent with previous studies[33], PT cells represented ~50% of the whole kidney mRNA (51.1% in Tgfbr2$^{fl/fl}$ and 55.6% in γGT-Cre;Tgfbr2$^{fl/fl}$). Cell type proportion analysis and Fisher's exact test ($p < 0.05$ and OddsRatio) indicated possible tissue remodeling at baseline and upon injury (Fig. 1c). *S1* PT cells proportion was higher in uninjured γGT-Cre;Tgfbr2$^{fl/fl}$ compared to Tgfbr2$^{fl/fl}$

(OR = 2.499), whereas myofibroblast and stromal cells (*Myo/St. mixed*) were lower in uninjured γGT-Cre;Tgfbr2$^{fl/fl}$ (OR = 0.5172), implying the importance of TGF-β signaling in basal PT remodeling and interstitial cellularity. Expectedly, *S3* PT cells could not be detected in both genotypes upon injury; and recently identified S3 type 2 PT cells (*S3T2*) were decreased in γGT-Cre;Tgfbr2$^{fl/fl}$ (OR = 0.8)[32,34]. Renal injury triggered a higher proportion of macrophages and dendritic cells (*Macro/Dend.*) in γGT-Cre;Tgfbr2$^{fl/fl}$ (OR = 1.779 3 weeks and 2.803 6 weeks after AA). These data suggest that genetic inhibition of TGF-β signaling in the PT affects the remodeling/survival of S3/S3T2 cells and inflammatory response under AA injury. Spatially resolved transcript images showed the increase in PT injury and fibrosis markers (Havcr1, Vcam1, and Col1a1) in γGT-Cre;Tgfbr2$^{fl/fl}$ compared to Tgfbr2$^{fl/fl}$ 3 weeks after AA injury (Fig. 1d), a time point where fibrosis is not marked in histology. H&E staining of kidney slices confirmed increased tubular injury (atrophy, dilatation, and flattening) score and Kim-1 mRNA levels indicated increased PT injury in γGT-Cre;Tgfbr2$^{fl/fl}$ mice compared to their Tgfbr2$^{fl/fl}$ littermates 6 weeks after AA (Fig. 1e, f and Supplementary Fig. 3). FACS analysis revealed an increase of CD45+ cell infiltrate in γGT-Cre;Tgfbr2$^{fl/fl}$ mice compared to their Tgfbr2$^{fl/fl}$ littermates 6 weeks after AA injury (Fig. 1g), supporting the cell proportion analysis and suggesting increased inflammation in γGT-Cre;Tgfbr2$^{fl/fl}$ mice upon chronic injury. Sirius red staining of collagens indicated a significant increase of cortical fibrosis in γGT-Cre;Tgfbr2$^{fl/fl}$ mice compared to their Tgfbr2$^{fl/fl}$ littermate 6 weeks after injury (Fig. 1h, i). Consistently, γGT-Cre;Tgfbr2$^{fl/fl}$ mice showed elevated blood urea nitrogen (BUN) levels, implying severe impairment of renal function (Fig. 1j). Taken together, these data demonstrated the beneficial role of TGF-β signaling in PT's adaptive response to chronic injury, and brought out spatiotemporal insights in post-injury tubular and interstitial remodeling and fibrosis induction.

### Proximal tubule TβRII deletion worsens mitochondrial injury and function in CKD

To determine the mechanisms whereby deleting TβRII worsens PT response to chronic injury, we performed RNAseq on conditionally immortalized PT cells treated or not with H$_2$O$_2$ to mimic oxidative stress. Deletion of TβRII deregulated 3359 genes ($p < 0.01$ and fc ≥ 1.5) and over-representation analysis of differentially expressed genes (DEGs) using EnrichR identified *Mitochondrion* as the most affected cell component in TβRII$^{−/−}$ PT cells (Supplementary Fig. 4a–c), emphasizing the importance of TβRII in PT mitochondrial homeostasis. Treatment of TβRII$^{flox/flox}$ PT cells with H$_2$O$_2$ increased mitochondrial (mt)-genome encoded proteins (Supplementary Fig. 4d). Interestingly, TβRII deletion mimicked H$_2$O$_2$-induced mtDNA replication, suggesting a basal oxidative stress in TβRII$^{−/−}$ PT cells (Supplementary Fig. 4e, f). Thus, we injured mice with AA and analyzed mitochondrial structure and function in vivo using electron and multiphoton microscopy techniques, respectively. γGT-Cre;Tgfbr2$^{fl/fl}$ mice have decreased mitochondrial length and increased mitochondrial injury in their PT cells compared to their Tgfbr2$^{fl/fl}$ littermates 3 weeks after AA injury (Fig. 2a–c). In vivo assessment of mitochondrial energetics using tetramethylrhodamine, methyl ester (TMRM) indicated mitochondrial dysfunction in the PT from both genotypes 3 weeks after AA injury; however, deleting TβRII aggravated mitochondrial dysfunction (Fig. 2d–f). Oil red O staining of lipids showed increased cortical lipid deposition in γGT-Cre;Tgfbr2$^{fl/fl}$ mice compared to their Tgfbr2$^{fl/fl}$ littermates, mirroring mitochondrial dysfunction and possible fatty acid metabolism impairment in the absence of TβRII (Fig. 2g, h).

To determine how TGF-β signaling affects mitochondrial function in vitro, we analyzed the oxygen consumption rate (OCR) on PT cells. Treatment of TβRII$^{flox/flox}$ PT cells with various doses of TGF-β1 revealed dose-dependent effects on OCR and the ability to produce ATP using OXPHOS (Supplementary Fig. 5a–e). The lowest dose of TGF-β1 (0.5 ng/ml) increased OCR, 1–2 ng/ml had no significant effect;

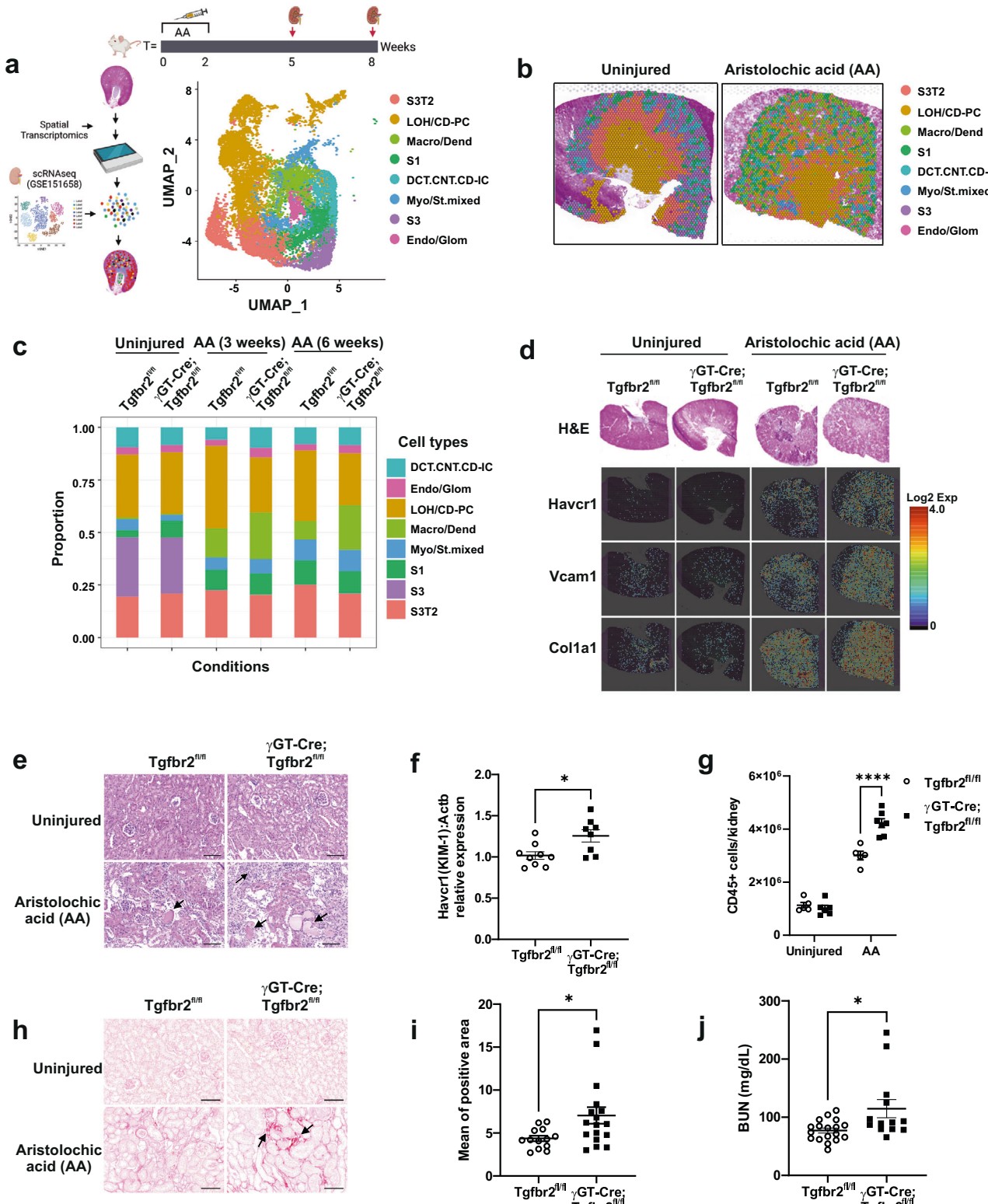

whereas higher doses (10 and 20 ng/ml) decreased OCR, implying that while some TGF-β signaling is necessary to maintain and promote homeostatic OXPHOS, excessive TGF-β signaling has detrimental effects on OCR and ATP production. Consistently, TβRII⁻/⁻ PT cells have decreased basal OCR and slightly increased extracellular acidification rate (ECAR) in line with decreased mitochondrial coupling efficiency (Supplementary Fig. 5f–l), suggesting that TGF-β signaling inhibition directly impacts the PT cell's capacity to efficiently produce ATP via OXPHOS. Fatty acid metabolism is pivotal for PTs to produce ATP

under normal conditions. Though PT cells become glycolytic in vitro, metabolic flux (Seahorse) analysis suggested that TβRII^flox/flox PT cells have kept a residual ability to metabolize palmitate, which is lost in TβRII⁻/⁻ PT cells (Supplementary Fig. 5m and Supplementary Fig. 6). Moreover, glucose preference analysis using a specific glycolysis inhibitor (UK5099) indicated increased glycolytic dependency in TβRII⁻/⁻ PT cells compared to TβRII^flox/flox PT cells (Supplementary Fig. 5n and Supplementary Fig. 7), implying a possible metabolic switch in the absence of TβRII. PT cells were not sensitive to amino acid

**Fig. 1 | Tubulo-interstitial remodeling and increased cortical injury and fibrosis in γGT-Cre;Tgfbr2$^{fl/fl}$ mice after AA injury. a** Schemes illustrating AA injury design, integrated data annotation strategy and renal cell types after dataset integration (UMAP plot). S3 type 2 PT cells (S3T2); Loop of Henle and principal cells (LOH/CD-PC); macrophages and dendritic cells (Macro/Dend.); S1 PT cells (S1); distal convoluted tubular, connecting and intercalated cells (DCT.CNT.CD-IC); myofibroblasts and stromal cells (Myo/St. mixed); S3 PT cells (S3); endothelial and glomerular cells (Endo/Glom). **b** Representative spatial plots of integrated clusters resolved on uninjured and injured kidney slices showing impaired cortico-medullary organization following injury. **c** Cell type proportions per conditions in the integrated data. **d** Representative H&E staining and spatial transcriptomics Cloupe browser kidney images with resolution of proximal tubule injury and fibrotic markers at baseline and 3 weeks after AA injury (one uninjured Tgfbr2$^{fl/fl}$ kidney, one uninjured γGT-Cre;Tgfbr2$^{fl/fl}$ kidney, one injured Tgfbr2$^{fl/fl}$ kidney and one injured γGT-Cre;Tgfbr2$^{fl/fl}$ kidney). Scale bars in source data. **e** Representative H&E images of kidneys from uninjured and 6 weeks AA injured mice. Arrows indicate injured tubules and interstitial cell infiltrate. **f** Relative Kim-1 mRNA levels in renal cortices 6 weeks after AA injury measured by RT-qPCR using Actb mRNA levels for normalization; $n = 9$ (Tgfbr2$^{fl/fl}$) and 8 (γGT-Cre;Tgfbr2$^{fl/fl}$) mice, $p = 0.014$. **g** Quantification of CD45+ cells from uninjured and injured renal leukocytes analyzed by FACS; $n = 6$ per genotypes of uninjured mice; $n = 5$ (Tgfbr2$^{fl/fl}$) and 7 (γGT-Cre;Tgfbr2$^{fl/fl}$) for injured mice, $p < 0.0001$. **h** Sirius red staining of kidneys from uninjured and AA injured mice showing collagen accumulation in red. Arrows indicate fibrotic areas. **i** Quantification of Sirius red positive area; $n = 13$ (Tgfbr2$^{fl/fl}$) and 17 (γGT-Cre;Tgfbr2$^{fl/fl}$) mice, $p = 0.0174$ (two-tailed Mann−Whitney test). **j** Plasma BUN levels measured 6 weeks after AA injury; $n = 18$ (Tgfbr2$^{fl/fl}$) and 13 (γGT-Cre;Tgfbr2$^{fl/fl}$) mice, $p = 0.0175$ (two-tailed Mann−Whitney test). All scale bars (**e** and **h**) represent 100 μm; dots represent the number of animals per group (**f, g, i** and **j**). Data are presented as mean values ± SEM. Statistical significance was determined by unpaired Student's $t$ test (two groups) or two-way ANOVA followed by Sidak's multiple comparisons test with $p < 0.05$ considered statistically significant unless otherwise stated. *$p < 0.05$; ****$p < 0.0001$. Source data are provided as a Source data file.

metabolism inhibitor (Supplementary Fig. 8a−e). To confirm TβRII deletion-induced metabolic switch in vitro, we measured ATP and lactate production using a bioluminescence assay. TβRII$^{−/−}$ PT cells have decreased ATP production and increased lactate production, suggesting a metabolic switch toward aerobic glycolysis (Warburg-like effect) in the absence of TβRII (Supplementary Fig. 5o, p). In line with the in vivo findings, AA treatment strikingly decreased OCR and the ability to produce ATP in TβRII$^{−/−}$ PT cells (Supplementary Fig. 9a−e). These results demonstrated the detrimental effects of TβRII deletion on PT mitochondrial structure and function in CKD.

### Deleting proximal tubule TβRII impairs mitochondrial complex I leading to oxidative stress

To understand the mechanism whereby TβRII deletion affects mitochondrial function, we performed pathway analysis on Visium PT-specific DEGs using Metacore. *Ubiquinone metabolism* was the most significantly affected pathway in γGT-Cre;Tgfbr2$^{fl/fl}$ PT cells compared to Tgfbr2$^{fl/fl}$ PT cells. The ubiquinone pathway map indicated basal downregulation of coenzyme Q2 and complex I subunits in γGT-Cre;Tgfbr2$^{fl/fl}$ PT (Fig. 3a, Supplementary Fig. 10a, b and Supplementary Table 4). Consistently, Ndufb8 protein expression was decreased in renal cortices of uninjured γGT-Cre;Tgfbr2$^{fl/fl}$ mice compared to their Tgfbr2$^{fl/fl}$ littermates (Fig. 3b). Cell fractionation confirmed the decrease of NDUFB8 in mitochondria of TβRII$^{−/−}$ PT cells (Fig. 3c), implying a possible regulatory effect of TGF-β signaling on complex I subunits. Moreover, TβRII$^{−/−}$ PT cells have decreased NAD+/NADH relative ratio as compared to TβRII$^{flox/flox}$ PT cells, suggesting complex I dysfunction in the absence of TβRII (Fig. 3d). Complex I dysfunction leads to ROS production and oxidative stress[35,36]. Assessment of ROS production using dichlorofluorescein (DCF) indicated basal increase of ROS production in TβRII$^{−/−}$ PT cells compared to TβRII$^{flox/flox}$ PT cells (Fig. 3e). To correlate increased ROS production with complex I expression in TβRII$^{−/−}$ PT cells, we treated PT cells with MitoQ, an antioxidant targeting mitochondria. Treatment of PT cells with MitoQ reduced basal ROS production in TβRII$^{−/−}$ PT cells to the level of TβRII$^{flox/flox}$ PT cells (Supplementary Fig. 11), implying the mitochondrial origin of increased ROS in TβRII$^{−/−}$ PT cells. Supplementation of PT cells with NAD+ significantly decreased ROS, improved ATP production, and decreased lactate production in TβRII$^{−/−}$ PT cells to the level of TβRII$^{flox/flox}$ PT cells (Fig. 3f−h), implying that deletion of TβRII induces mitochondrial dysfunction and metabolic switch, partly, through impaired expression and function of complex I subunits.

### Deleting proximal tubule TβRII impairs the mitochondrial quality control

Mitochondria quality control is mainly mediated by their renewal through biogenesis and mitophagy. Pgc1α, the master regulator of mitochondrial biogenesis, has been shown to be protective in mouse models of CKD[37–39], though it is negatively regulated by TGF-β[40,41]. To investigate how TGF-β affects Pgc1α in PT cells, we treated TβRII$^{flox/flox}$ PT cells with various doses of TGF-β1 and analyzed Pgc1α mRNA and protein expression (Supplementary Fig. 12). TGF-β1 decreased Pgc1α mRNA, and transient phosphorylation of Smad3 correlated with downregulation of Pgc1α protein, suggesting that supra-optimal TGF-β doses repress Pgc1α expression in PT cells (Supplementary Fig. 12a, b). BulkRNAseq pathway analysis using Metacore revealed "*Oxidative stress and PGC-1alpha in activation of antioxidant defense system*" as the two most affected pathways in TβRII$^{−/−}$ PT cells (Supplementary Fig. 12c). Transcript levels of Pgc1α and Tfam, a mitochondrial biogenesis transcription factor, were increased in TβRII$^{−/−}$ PT cells compared to TβRII$^{flox/flox}$ PT cells (Supplementary Fig. 12d, e). Immunoblot analysis confirmed the increase of Pgc1α in TβRII$^{−/−}$ PT cells (Supplementary Fig. 12f), in line with the increase of mtDNA copy number in TβRII$^{−/−}$ PT cells. Unexpectedly, DNA polymerase γ, the only polymerase involved in mtDNA replication, was decreased in TβRII$^{−/−}$ PT cells (Supplementary Fig. 12g), implying a possible mt-genomic instability in TβRII$^{−/−}$ PT cells. Consistently, uninjured γGT-Cre;Tgfbr2$^{fl/fl}$ mice have increased cortical Pgc1α mRNA and activation (nuclear translocation) compared to their Tgfbr2$^{fl/fl}$ littermates (Fig. 4a−c), suggesting that TGF-β signaling modulates Pgc1α activation and mitochondrial content in PT. Pgc1α mRNA was significantly decreased 6 weeks after AA injury in γGT-Cre;Tgfbr2$^{fl/fl}$ mice, but not in Tgfbr2$^{fl/fl}$ mice. In contrast to mRNA level, Pgc1α activation was decreased and not significantly different between genotypes 6 weeks after AA injury.

To analyze mitophagy, we crossed our mice with the mito-QC reporter mouse which expresses a pH-sensitive tandem mCherry-GFP tag fused to a fragment (residues 101–152) of the mitochondrial protein Fis1. Under mitophagy, GFP is quenched in the lysosomes and only mCherry is expressed[42]. Six weeks after AA injury, γGT-Cre;Tgfbr2$^{fl/wt}$;mito-QC mice have decreased mitophagy as compared to their Tgfbr2$^{fl/fl}$ littermates (Fig. 4d, e). Consistently, the central autophagic protein LC3A-I is decreased in γGT-Cre;Tgfbr2$^{fl/fl}$ mice as compared to their floxed littermates 6 weeks after AA injury (Fig. 4f−h). Moreover, Pink1, an important mediator of mitophagy, was decreased in TβRII$^{−/−}$ PT cells (Supplementary Fig. 12h). Altogether, these data suggest a central modulating role of TGF-β signaling in PT mitochondrial quality control.

### Deleting proximal tubule TβRII worsens Th1 inflammatory response under AA-induced injury

To delineate how TGF-β signaling affects PT-immune cell interactions in CKD, we injured mice and profiled renal inflammation using Visium and flow cytometry (FACS). Trajectory analysis in the *Myo/St. mixed* clusters revealed increased macrophage

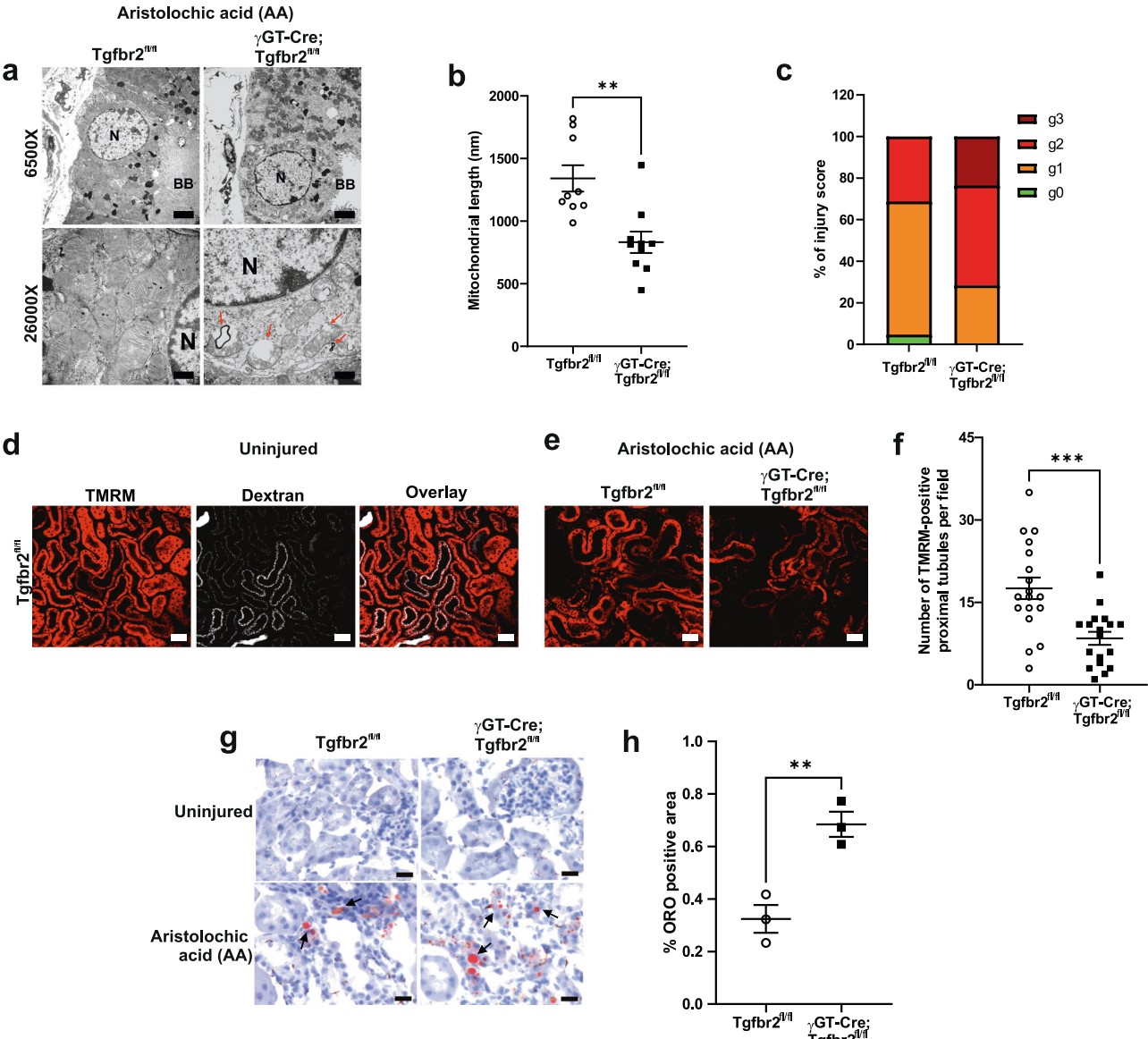

**Fig. 2 | Increased mitochondrial injury and dysfunction in γGT-Cre;Tgfbr2^fl/fl mice 3 weeks after AA injury. a** Representative transmission electron microscopy images showing increased mitochondrial injury (decreased cristae number, increased vacuolization, and myelin figures) in the PT from γGT-Cre;Tgfbr2^fl/fl kidneys. Scale bars represent 500 nm; N = nucleus; BB = brush border; arrows highlight vacuolated mitochondria and presence of myelin figures.
**b**, **c** Quantification of mitochondrial length ($n = 9$ different areas from 2 different animals per group, $p = 0.0014$) and injury score (g0: no injury, g1: slight decrease of cristae number or small vacuoles; g2: severe decrease of cristae or big vacuoles formation; g3: severe decrease of cristae, big vacuoles formation and myelin figures) in proximal tubules. The dots represent the mean of PT mitochondria length in the representative cortical area per group. **d** Representative multiphoton live microscopy images illustrating PT dextran uptake and TMRM incorporation in mitochondria from uninjured kidneys to respectively assess PT absorptive function and mitochondrial membrane potential. Scale bars represent 50 μm. Two

groups (4 animals/8 kidneys) were analyzed in two independent experiments. **e** Representative TMRM images showing decreased mitochondrial membrane potential in the PTs from injured γGT-Cre;Tgfbr2^fl/fl mice. Scale bars represent 50 μm. **f** Quantification of TMRM in injured PTs. Every dot represents the number of TMRM positive tubules in the field of 900 μm² (in one capture); $n = 2$ (Tgfbr2^fl/fl) and 2 (γGT-Cre;Tgfbr2^fl/fl) mice, $p = 0.0004$. **g** Representative image of oil red O staining showing increased lipids deposition (red) in the renal cortex of γGT-Cre;Tgfbr2^fl/fl mice. Scale bars represent 50 μm. **h** Quantification of oil red O positive area in the renal cortex of injured mice; $n = 3$ (Tgfbr2^fl/fl) and 3 (γGT-Cre;Tgfbr2^fl/fl) mice, $p = 0.0073$. The dots represent the number of animals per group in oil red O staining. Data are presented as mean values ± SEM. Statistical significance was determined by unpaired Student's $t$ test (two groups) with $p < 0.05$ considered statistically significant. **$p < 0.01$; ***$p < 0.001$. Source data are provided as a Source data file.

markers in injured γGT-Cre;Tgfbr2^fl/fl dataset as compared to Tgfbr2^fl/fl notably 3 weeks after AA injury (Supplementary Fig. 13a). Cell–cell communication analysis using the Cellchat database of Ligand/Receptor (LR) pairs indicated that γGT-Cre;Tgfbr2^fl/fl *S3T2* cells have increased interactions with *Macro/Dend* cells compared to Tgfbr2^fl/fl *S3T2* cells 3 weeks after AA injury (Supplementary Fig. 13b). Immuno-fluorescence microscopy confirmed increased F4/80+ cell infiltrate in γGT-

Cre;Tgfbr2^fl/fl renal cortices compared to Tgfbr2^fl/fl mice 3 weeks after AA injury (Supplementary Fig. 13c). FACS analysis showed increased number of dendritic cells in γGT-Cre;Tgfbr2^fl/fl mice in line with cell–cell communication data (Supplementary Figs. 13d and 14). These results suggest that injured PT cell/macrophage/dendritic cell crosstalk is a major component in the PT's response to late acute injury (3 weeks). Analysis of factors involved in *S3T2-Macro/Dend.* identified possible adaptive/maladaptive interacting

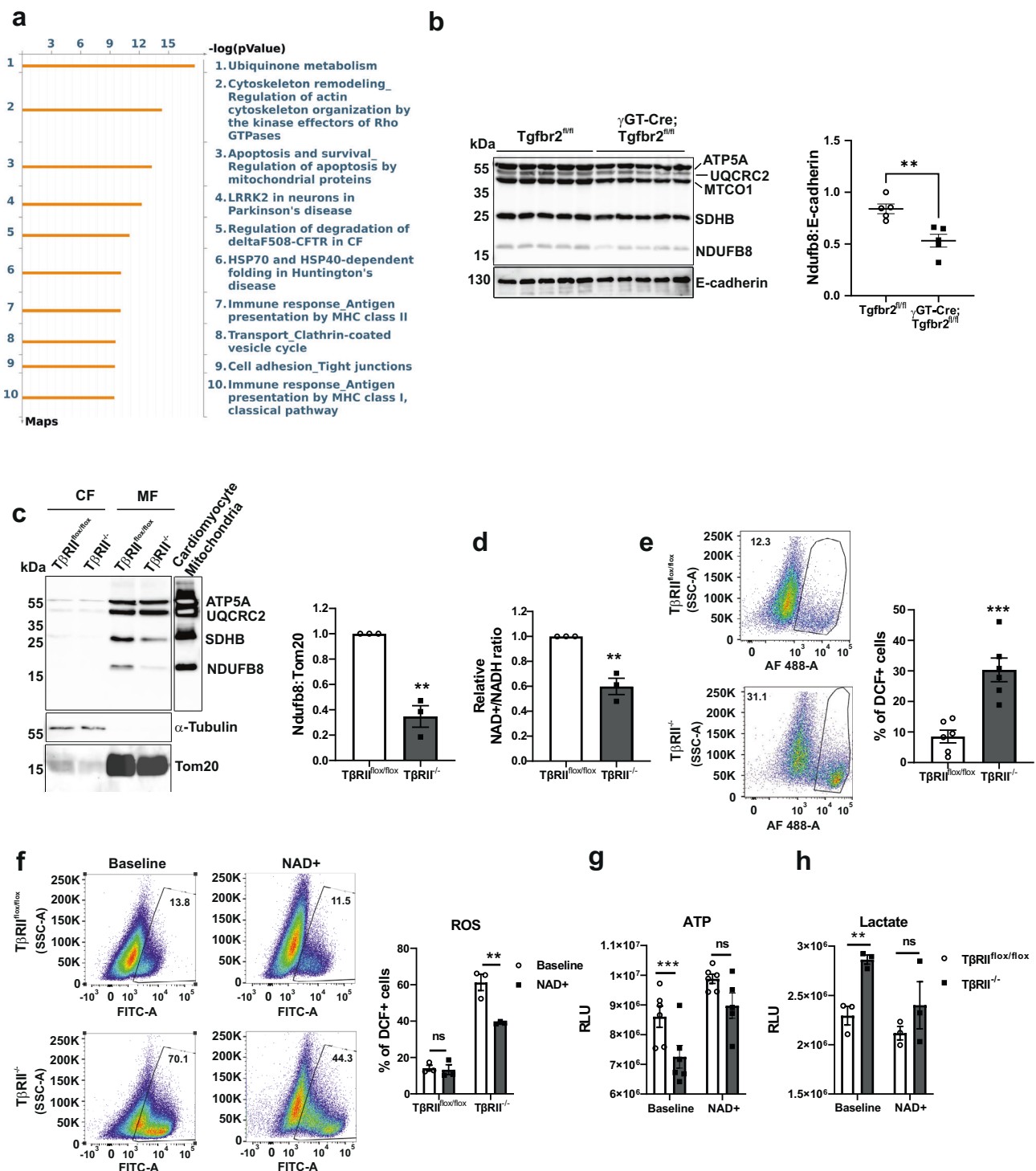

factors among which the colony-stimulating factor (Csf) and angiopoietin-like protein (Angptl) showed restricted *S3T2-Macro/Dend.* interaction in Tgfbr2[fl/fl] and in γGT-Cre;Tgfbr2[fl/fl] respectively (Supplementary Fig. 15a). Normalized RNAseq signals of these factors corroborated the in vivo findings. Angptl7/8 signals were strikingly increased whereas Angptl2/4/6 were decreased in TβRII[−/−] PT cells. Csf1 and Csfr1 were consistently decreased in TβRII[−/−] PT cells whereas Csf2r, a receptor involved in T cell activation, was increased in TβRII[−/−] PT cells compared to TβRII[flox/flox] PT cells. Finally, Notch target genes and other possible adaptive factors were decreased in TβRII[−/−] PT cells as compared to TβRII[flox/flox] PT cells (Supplementary Figs. 15b and 16).

Furthermore, sub-clustering analysis of *Myo/St. mixed* cluster revealed the presence of both myofibroblast and immune cells including cells expressing B cell's markers (Supplementary Fig. 17). However, FACS analysis did not show differences in B cells between injured genotypes (Supplementary Fig. 18).

Mitochondrial damage-associated molecular patterns (DAMPS) reportedly lead to the activation of Cgas/Sting/IFNγ axis in renal fibrosis[43,44]. Pathway analysis of DEGs in 6 weeks injured *S3T2* clusters using Metacore indicated innate immunity (complement system and IFNγ) among the top 10 significantly affected pathways in γGT-Cre;Tgfbr2[fl/fl] *S3T2* (Fig. 5a). Though macrophages and dendritic cells initiate inflammation in response to tubular injury, T cells are involved

**Fig. 3 | Proximal tubule TβRII deletion disrupts mitochondrial complex I leading to oxidative stress and a metabolic rewiring. a** Metacore pathway analysis of differentially expressed genes in uninjured PT clusters showing the top 10 significantly affected pathways in γGT-Cre;Tgfbr2^fl/fl compared to Tgfbr2^fl/fl. **b** Immunoblotting of OXPHOS proteins showing a significant decrease of complex I (NDUFB8) expression in uninjured renal cortices of γGT-Cre;Tgfbr2^fl/fl mice compared to their Tgfbr2^fl/fl littermates; $n = 5$ (Tgfbr2^fl/fl) and 5 (γGT-Cre;Tgfbr2^fl/fl) mice, $p = 0.0041$. E-cadherin is used as marker of renal parenchyma and loading control. The dots represent the number of animals per group. **c** Cell fractionation followed by immunoblotting and quantification of OXPHOS proteins showing a significant decrease of complex I (NDUFB8) basal expression in mitochondria of TβRII^−/− PT cells ($n = 3$ independent experiments, $p = 0.0015$). **d** Bioluminescence measurement of NAD+/NADH ratios showing a decreased relative ratio in TβRII^−/− PT cells ($n = 3$ independent biological replicates, $p = 0.0036$). **e** FACS analysis of

DCF-positive cells showing increased basal ROS production in TβRII^−/− PT cells ($n = 6$ independent experiments, $p = 0.0006$). **f** NAD+ treatment significantly decreased ROS in TβRII^−/− PT cells, assessed with DCF and measured by FACS ($n = 3$ independent biological replicates, $p = 0.0027$). **g** NAD+ treatment increased ATP production in TβRII^−/− PT cells to the same level as in TβRII^flox/flox PT cells, measured by a bioluminescence assay ($n = 6$ independent biological replicates, baseline $p = 0.0001$ and NAD+ treatment $p = 0.4025$). **h** NAD+ treatment decreased lactate production in TβRII^−/− PT cells to the level of TβRII^flox/flox PT cells, measured by a bioluminescence assay ($n = 3$ independent biological replicates, baseline $p = 0.0057$ and NAD+ treatment $p = 0.3175$). Data are presented as mean values ± SEM. Statistical significance was determined by unpaired Student's $t$ test (two groups) or two-way ANOVA followed by Sidak's multiple comparisons test with $p < 0.05$ considered statistically significant. *$p < 0.05$; **$p < 0.01$; ****$p < 0.0001$. Source data are provided as a Source data file.

in the whole evolution of injury[45]. We investigated the Th1 inflammatory response 6 weeks after AA injury. CD3+ cell infiltrate was not significantly different between genotypes in IHC (Supplementary Fig. 19a); however, FACS analyses revealed increased CD4+ cells in γGT-Cre;Tgfbr2^fl/fl kidneys compared to Tgfbr2^fl/fl kidneys. CD8+ cells were augmented in γGT-Cre;Tgfbr2^fl/fl kidneys, but not significantly different at this time point (Fig. 5b–d and Supplementary Fig. 14). IFNγ and TNFα + CD4 but not CD8 cells, were significantly increased in γGT-Cre;Tgfbr2^fl/fl compared to Tgfbr2^fl/fl kidneys (Fig. 5e–h). Moreover, the percentage of the reno-protective Foxp3+ (T reg) CD4+ cells out of CD45+ cells was decreased in γGT-Cre;Tgfbr2^fl/fl mice (Supplementary Fig. 19b, c). Cgas and Sting protein levels were increased in γGT-Cre;Tgfbr2^fl/fl compared to Tgfbr2^fl/fl renal cortices (Fig. 5i–k), suggesting that TβRII deletion-induced mitochondrial dysfunction worsens Th1 inflammatory response in acute to chronic PT injury.

## TGF-β receptors are decreased in CKD patients

Excessive TGF-β arguably promotes fibrosis in CKD. However, targeting TGF-β signaling inhibition failed to mitigate fibrosis in humans[46,47], indicating that TGF-β is not the mainstay component in renal fibrosis which involves concerted activation of multiple factors. We therefore analyzed PT expression of TGF-β receptors (Tgfbr1, Tgfbr2, and Tgfbr3) in a snRNAseq database of healthy and CKD (eGFR<60 ml/min) kidney biopsies[48]. Surprisingly, TGF-β receptors are significantly decreased whereas potential maladaptive factors identified in this study were increased (Supplementary Fig. 20) in the PT of CKD patients as compared to healthy controls, implying the beneficial effect of intact/physiological TGF-β signaling in PT response to CKD (Fig. 6a–d). In accordance with our findings in mice, pathway activity analysis confirmed impaired mitochondrial homeostasis and metabolism, notably complex I activity in the PT of CKD patients (Fig. 6e). Taken together, these results suggest a beneficial role of intrinsic TGF-β signaling in PT response to CKD in humans.

## Discussion

The PT is the most vulnerable renal segment, and the injured PT plays a pivotal role in the pathogenesis of CKD. We previously reported that deleting TβRII in the PT worsens renal tubular injury and cortical fibrosis in two mouse models of CKD. We hereby spatiotemporally delineate the beneficial role of TGF-β signaling in PT response to CKD using Visium and biochemical approaches. Our data spatially demonstrate that having intact TGF-β signaling in PTs promotes an adaptive response to CKD and show that TβRII deletion in the PT worsens mitochondrial injury and Th1 immune response in CKD. Despite the low resolution of Visium and the use of endotoxemia scRNAseq reference database, cluster annotation identified the main renal cell types with proportions of PT cells similar to prior reports[33]. The use of a different injury model scRNAseq as reference may affect cell proportion; however, additional data curating and quality control analyses were performed to ensure the accuracy of cell type identification. Further investigations using AA

scRNAseq as reference dataset and renal segment functional analysis (urine osmolarity and concentration ability after water loading and subsequent deprivation) will help to understand the effect of TGF-β signaling on tubular and interstitial remodeling. We found that S1 and S3T2 PT cells mediate differential interactions with tubular and interstitial compartments in injury and identified injured γGT-Cre;Tgfbr2^fl/fl S3T2 rather than S1 cells as the main cells interacting with macrophages and dendritic cells in the late acute injury phase. Previous studies reported the existence of S3T2 PT cells which express specific markers including angiotensinogen, Rnf24, Atp11a, Scl22a7, and Slc22a13, and dwell in the outer stripe of the outer medulla[32,34]. The difference between S3 and S3T2 cells may come from their respective localization. The S3T2 cells are seemingly more resistant to injury-induced cell death or trans-differentiation than the cortical S3 cells. Collagen1a1 transcript, a marker of fibrosis, was enhanced in γGT-Cre;Tgfbr2^fl/fl mice 3 weeks after AA. Though fibrosis was less pronounced at 3 weeks compared to 6 weeks, future investigations should consider earlier time points (1 day and 1 week after AA injections) to better elucidate the kinetics of the events preceding fibrosis.

TβRII deletion impairs OXPHOS and leads to a metabolic switch. Our data corroborate previous studies where stimulation of podocytes with TGF-β increased OCR and oxidative stress[49]. TGF-β1 reportedly increases or decreases glycolytic enzymes depending on the cell types[50]. The effect of NAD+ on ROS production, ATP and lactate production suggests that exacerbated aerobic glycolysis dependency is subsequent to mitochondrial dysfunction. TGF-β has been reported to reduce mitochondrial complex IV in lung epithelial cells and regulate mitochondrial UCP2 in breast tumor cells expression[51,52]. PT deletion of TβRII decreased the expression of polymerase γ, the only polymerase involved in the replication of mtDNA[53], implying possible mt-genome instability in the absence of TGF-β signaling. Moreover, MTCO1 (complex IV) tended to decrease in uninjured γGT-Cre;Tgfbr2^fl/fl compared to their Tgfbr2^fl/fl littermates, though statistical significance was not reached.

Our data showed that TβRII deletion in PT cells induced Pgc1α expression and mitochondrial biogenesis which is in line with studies reporting negative regulation of Pgc1α expression by TGF-β signaling[40,41,54]. However, TβRII deletion-induced Pgc1α activation in PT is associated with oxidative stress and susceptibility to AA-induced injury, opposing previously reported protective effects of Pgc1α in renal fibrosis[37–39,55]. This apparent discrepancy can be explained by the possible primacy of TβRII deletion-induced ETC dysfunction on mitochondrial biogenesis. Indeed, TβRII deletion may primarily affect ETC integrity due to a cell reprogramming mechanism or decreased Polγ, which likely keep a permanent harmful pressure on mitochondrial-genome and compromise the beneficial effect of subsequent Pgc1α activation. Under such pre-existing mitochondrial dysfunction, Pgc1α activation becomes maladaptive by amplifying mitochondrial dysfunction and oxidative stress. Pgc1α has been reported to ameliorate muscle function during aging[56]. However, in

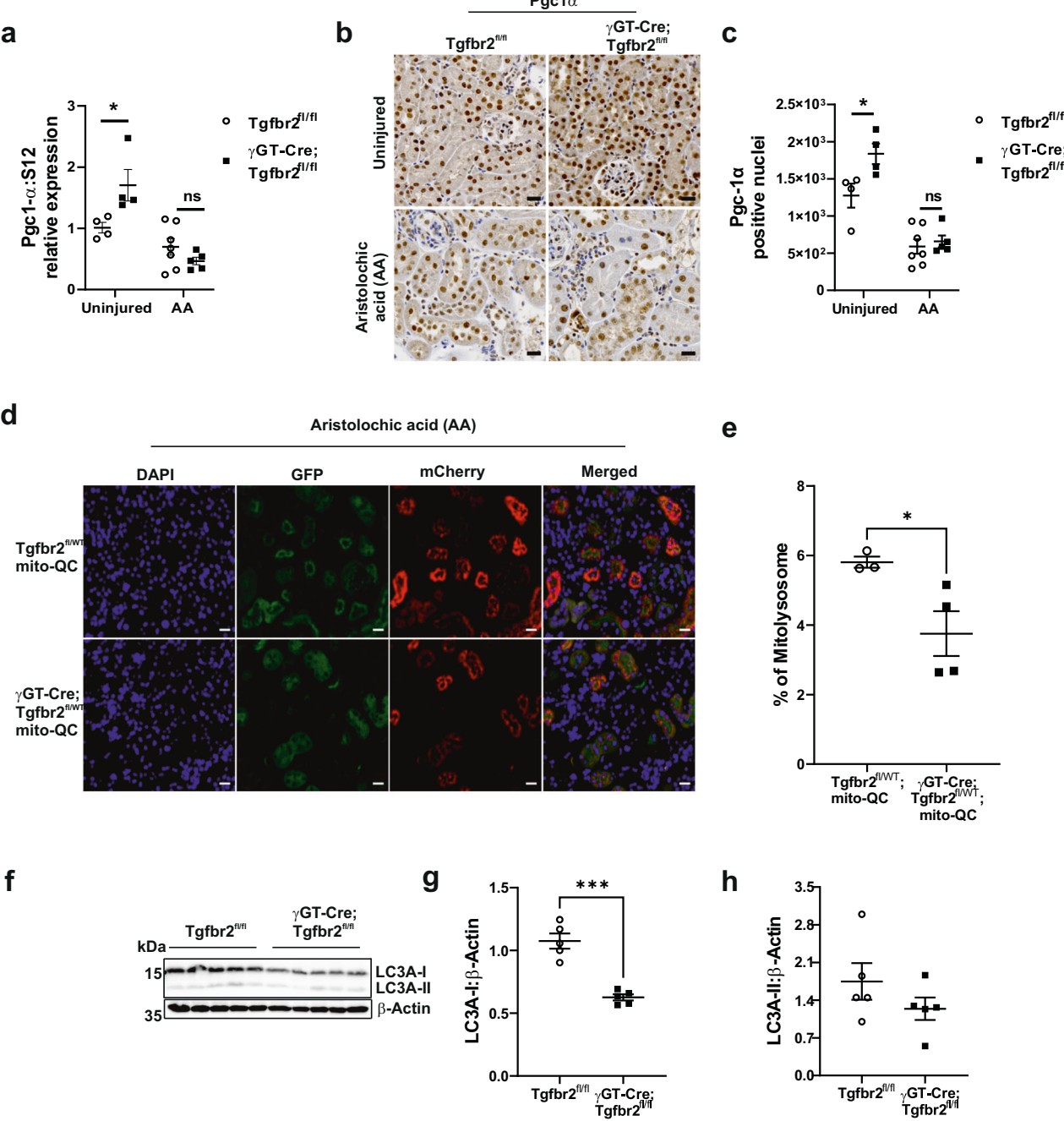

**Fig. 4 | Proximal tubule TβRII deletion impairs mitochondrial quality control.**
**a** Relative Pgc1α mRNA levels in renal cortices from uninjured and AA injured (6 weeks) mice, measured by RT-qPCR using S12 as housekeeping control gene; $n = 4$ uninjured, 7 injured (Tgfbr2[fl/fl]) and 4 uninjured, 5 injured (γGT-Cre;Tgfbr2[fl/fl]) mice, uninjured $p = 0.05$ and injured $p = 0.8152$.
**b**, **c** Representative Pgc1α immunohistochemistry (IHC) images and quantification in uninjured and 6 weeks injured renal cortices; $n = 4$ uninjured, 7 injured (Tgfbr2[fl/fl]) and 4 uninjured, 5 injured (γGT-Cre;Tgfbr2[fl/fl]) mice, uninjured $p = 0.0436$ and injured $p = 0.9983$. Scale bars represent 20 μm.
**d**, **e** Representative endogenous fluorescence images and quantification showing decreased mitophagic flux (mCherry) in γGT-Cre;Tgfbr2[fl/WT];mito-QC

mice compared to Tgfbr2[fl/WT];mito-QC mice 6 weeks after AA injury; $n = 3$ (Tgfbr2[fl/WT];mito-QC) and 4 (γGT-Cre;Tgfbr2[fl/WT];mito-QC) mice, $p = 0.045$. Scale bars represent 20 μm. **f**–**h** LC3A (I/II) immunoblotting and quantification showing significant decrease of LC3A-I expression in γGT-Cre;Tgfbr2[fl/fl] renal cortices 6 weeks after AA injury; $n = 5$ (Tgfbr2[fl/fl]) and 5 (γGT-Cre;Tgfbr2[fl/fl]) mice, $p = 0.0001$. Data are presented as mean values ± SEM. Statistical significance was determined by unpaired Student's $t$ test (two groups) or two-way ANOVA followed by Sidak's multiple comparisons test, with $p < 0.05$ considered statistically significant. The dots represent the number of animals per group. *$p < 0.05$; ***$p < 0.001$. Source data are provided as a Source data file.

aging the mitochondrial quality control efficiency is decreased and cells are composed of a mixture of low and high-quality mitochondria, so that Pgc1α activation shows a beneficial net effect. Our data also indicate impaired mitophagy in γGT-Cre;Tgfbr2[fl/fl] mice, suggesting that conditional knockout mice accumulate damaged mitochondria

under chronic injury, which likely amplifies the pre-existing mitochondrial dysfunction created by TβRII deletion.. This result is supported by previous studies where TGF-β signaling promoted autophagy/mitophagy and mediated mitochondrial elongation in ARPE-19 cells through downregulation of Opa3[57].

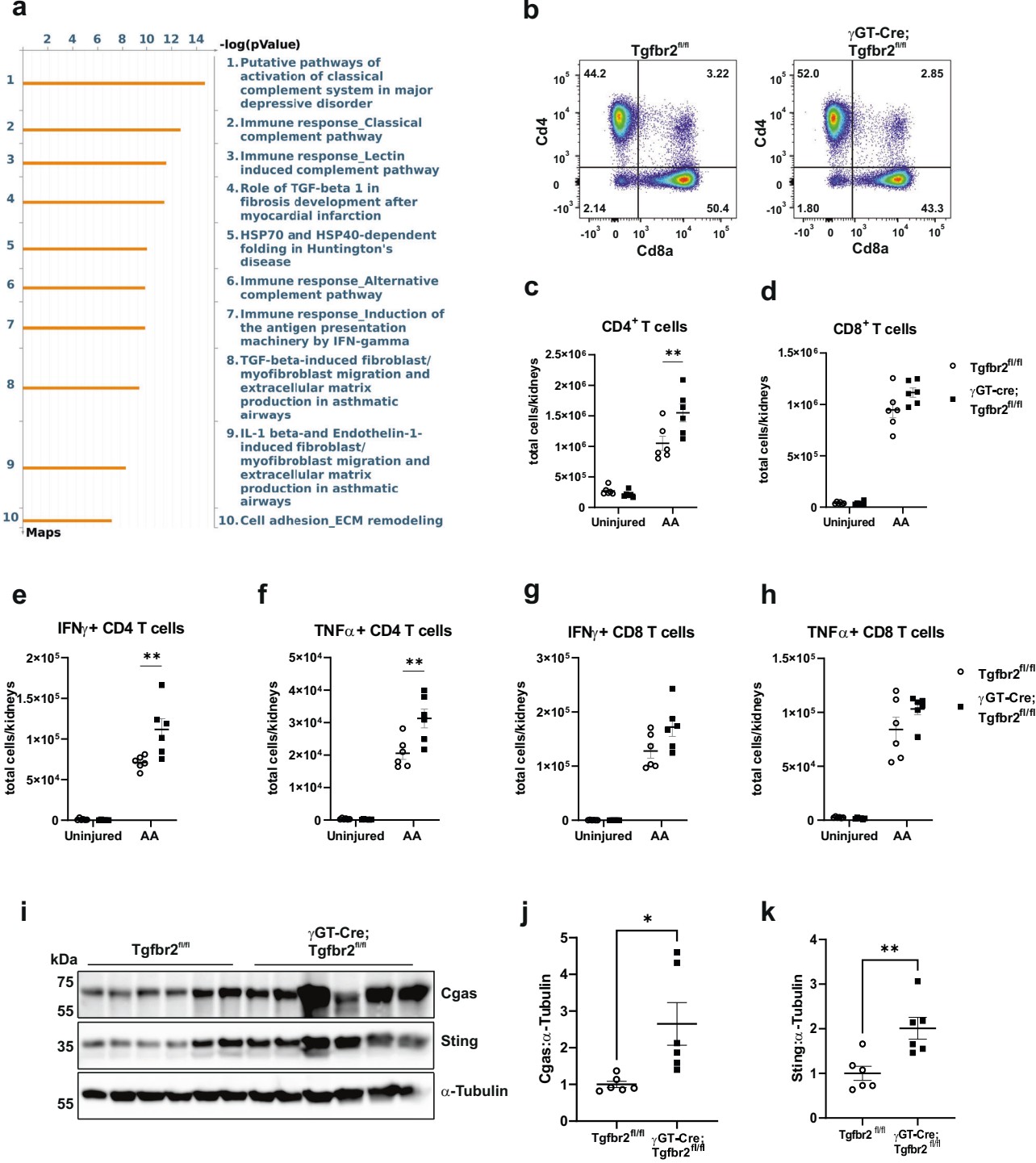

**Fig. 5 | Proximal tubule TβRII deletion increases the Th1 inflammatory response 6 weeks after AA injury. a** Metacore pathway analysis of differentially expressed genes in injured PT clusters showing the top 10 significantly affected pathways in γGT-Cre;Tgfbr2$^{fl/fl}$ compared to Tgfbr2$^{fl/fl}$ 6 weeks after AA injury. **b–d** FACS analyses of renal leukocytes showing significant increase of CD4 + T cell number in kidneys of γGT-Cre;Tgfbr2$^{fl/fl}$ mice compared to those from their Tgfbr2$^{fl/fl}$ littermates 6 weeks after AA injury; $n = 6$ (Tgfbr2$^{fl/fl}$) and 6 (γGT-Cre;Tgfbr2$^{fl/fl}$) mice, $p = 0.0085$. CD8+ T cell numbers were increased in injured Tgfbr2$^{fl/fl}$ compared to γGT-Cre;Tgfbr2$^{fl/fl}$ kidneys, but did not reach statistical significance; $n = 6$ (Tgfbr2$^{fl/fl}$) and 6 (γGT-Cre;Tgfbr2$^{fl/fl}$) mice, $p = 0.0984$. **e–h** FACS analyses of renal leukocytes showing a significant increase of the number of IFNγ and TNFα producing CD4+ T cells in kidneys of γGT-Cre;Tgfbr2$^{fl/fl}$ mice compared to those from their Tgfbr2$^{fl/fl}$ littermates 6 weeks after AA injury; $n = 6$ (Tgfbr2$^{fl/fl}$) and 6 (γGT-

Cre;Tgfbr2$^{fl/fl}$) mice, $p = 0.0027$ (IFNγ)and $p = 0.0016$ (TNFα). IFNγ and TNFα producing CD8 + T cell numbers were not statistically different between genotypes; $n = 6$ (Tgfbr2$^{fl/fl}$) and 6 (γGT-Cre;Tgfbr2$^{fl/fl}$) mice, $p = 0.0519$ (IFNγ) and $p = 0.2419$ (TNFα). **i–k** Cgas/Sting immunoblotting and quantification showing significantly increased expressions in renal cortices of γGT-Cre;Tgfbr2$^{fl/fl}$ mice compared to those from their Tgfbr2$^{fl/fl}$ littermates 6 weeks after AA injury; $n = 6$ (Tgfbr2$^{fl/fl}$) and 6 (γGT-Cre;Tgfbr2$^{fl/fl}$) mice, Cgas $p = 0.185$ and Sting $p = 0.0061$. α-Tubulin was used as loading and blotting control. Data are presented as mean values ± SEM. Statistical significance was determined by unpaired Student's $t$ test (two groups) or two-way ANOVA followed by Sidak's multiple comparisons test, with $p < 0.05$ considered statistically significant. The dots represent the number of animals per group. *$p < 0.05$; **$p < 0.01$. Source data are provided as a Source data file.

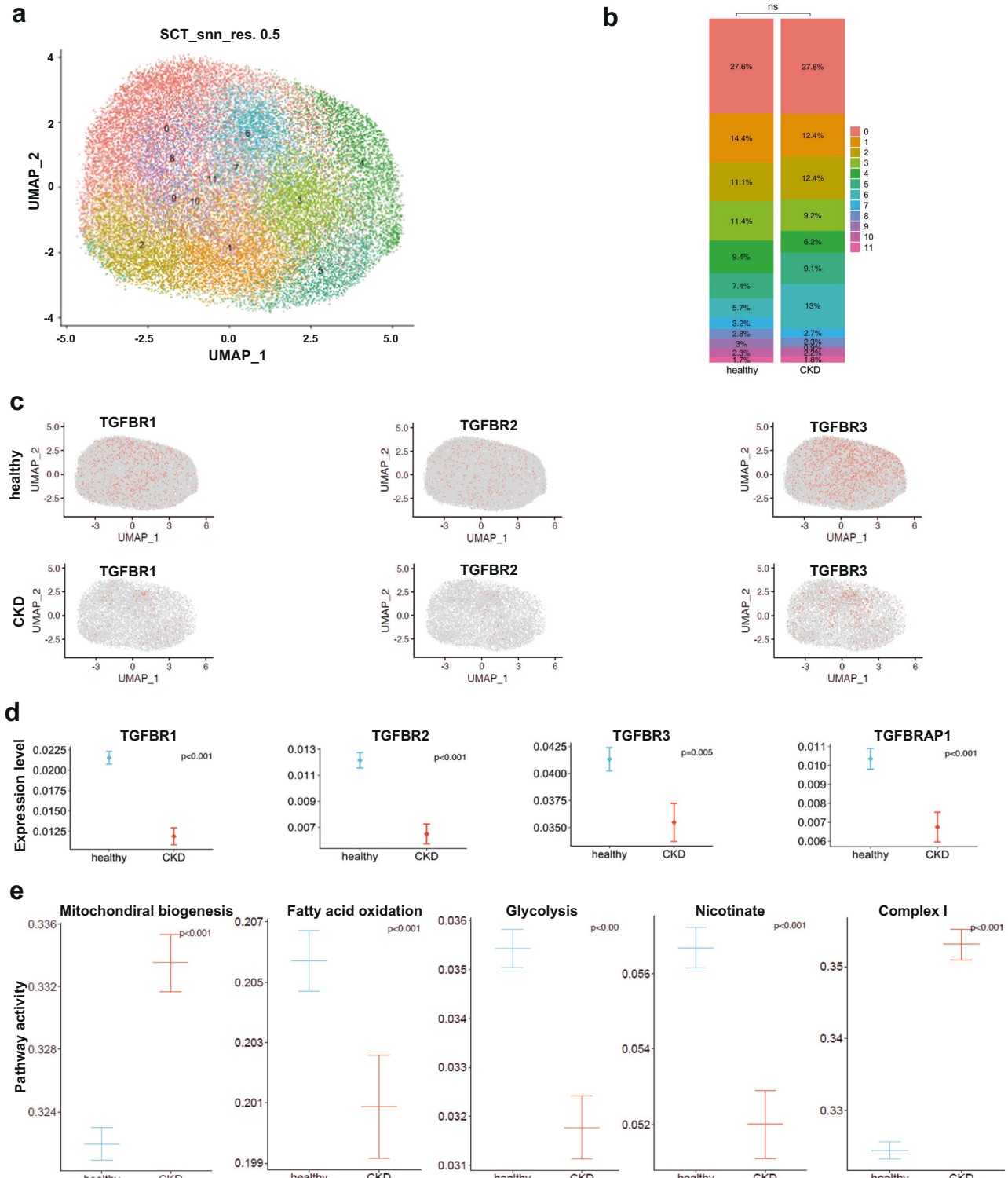

**Fig. 6 | Decreased expression of TGF-β receptors and impaired metabolism in the proximal tubule of CKD patients. a** UMAP plot of normalized data clustering. Numbers represent PT clusters. **b** Bar plot depicting the proportion of PT cells per clusters in healthy and CKD kidney biopsy datasets. **c** UMAPs showing TGFBRs (1, 2, and 3) features in healthy and CKD PT cells. **d** Differential gene expression analysis showing a significant decrease of TGFBRs (1, 2, and 3) and TGFBRAP1 in CKD PT cells as compared to healthy PT cells. **e** Pathway activity analysis showing impaired mitochondrial biogenesis, complex I activity, and metabolism. $N = 5$ CKD patients (eGFR<60) and 3 healthy controls. Differential gene expression was evaluated using the Wilcoxon Rank Sum test from the *FindMarkers* function. Calculation of the pathway scores was performed using REACTOME genesets.

TGF-β arguably modulates inflammation in CKD. Global knockout of TGF-β1 or TβRII induce lethal inflammatory disorders in mice[20–23]. We demonstrated that PT-specific deletion of TβRII worsens Th1 inflammatory response in chronic injury, probably, through an aberrant activation of macrophage and dendritic cells by injured *S3T2* PT cells involving multiple LR pairs. A limitation of our vision analysis is a lack of deconvolution methods which would facilitate co-localization of immune cell infiltrate and epithelial cells as shown in histology. Based on our data and the literature, Csf, Egf, ncWnt, and Notch may mediate adaptive *S3T2-Macro/Dend* interactions; whereas, Angptl, Fgf, Igf, and Cxcl may be maladaptive mediators. Csf1 mitigates tubular injury following ischemia/reperfusion; whereas Angptl-2 crosstalks with TGF-β to promote fibrosis in human CKD[58,59]. Though we did not establish a causative relationship between mitochondrial injury and inflammation; our findings are in line with previous studies whereby DAMPS specifically activate the Cgas/Sting/IFNγ axis. Moreover, the decrease of the reno-protective Treg cells in injured γGT-Cre;Tgfbr2[fl/fl] mice is consistent with a beneficial effect of intact TGF-β signaling in the PT's response to inflammation and chronic injury.

Decreased TβRII in the PT (specifically in cortical CD10+ cells) of CKD patients substantiates the beneficial effect of TGF-β signaling in PT response to human CKD. Taken together with our mouse data, these findings imply a beneficial role of epithelial TGF-β signaling in PT cell homeostasis and metabolism.

In conclusion, we show that deleting TβRII in the PT worsens mitochondrial injury and Th1 inflammatory response in the AA model of CKD. The mechanism includes impaired complex I expression, compromised effect of Pgc1α activation and impaired mitophagy, and aberrant *S3T2-Macrophage/Dendritic* cell interactions. Although previous studies have explored the kinetics of immune response in renal injury, this study clearly identified cellular and molecular contributors in the PT's maladaptive response to chronic injury that might be targeted to hamper CKD progression in humans.

## Methods

### Animal models

All procedures were approved by the veterinary office of the canton Zurich, Switzerland (ZH123/19). Prior to experimental commitment, mice were tagged using ear notching in accordance with the Laboratory Animal Services Center (LASC) license 101, and generated tissue were used for genotyping. After genotyping, animals were transferred to the experimental room, where they were allowed to acclimate for at least 7 days before starting experiments. During the acclimation period, mice were randomly assigned as controls or CKD and monitored to ensure water and food ad libitum accessibility every other day. Given the pathocentric strain of CKD models, the use of painkillers is mandatory to minimize procedure-related pain. Mice were monitored every other day during 2 weeks of intraperitoneal AA injections (six injections every other day), then every day in the acute phase (7 days after the last AA injection), and finally every other day afterward until the experiment endpoint. Mice were scored for signs of pain (hunched posture, poor grooming, reduced mobility, and subsequent body weight loss) every day in the acute phase. According to our pain scoring criteria, mice were provided wet food pellets and/or administered pre-warmed Ringer's lactate/5% glucose solution and/or buprenorphine (0.1 mg/kg) diluted in 0.9% NaCl (1 ml of 0.3 mg/ml of buprenorphine in 5 ml of 0.9% NaCl to have 2 microliters/g body weight). Buprenorphine is an opioid and strong analgesic that we preferred in this study because of its long-lasting effect (6–8 h), and compared to other opioids (butorphanol for instance), it reportedly has minimal hemodynamic side effects which is a very important aspect in this study. Euthanasia was considered in case of failure of pain mitigating measures and at the experimental endpoint. If euthanasia is needed before the experimental endpoint, mice were submitted to 70% $CO_2$ and only the kidneys were collected for further investigation. In case we cannot decrease animal distress in the assigned time period, concerned animal was immediately euthanized and the kidneys were collected for further investigation. At the experiment endpoint, mice were anesthetized by inhalation of 5% isoflurane in oxygen as carrier gas using the VetFlo stand. After confirmation of complete anesthesia by checking the pedal withdrawal reflex and tail pinch three times, mice were euthanized by cervical dislocation. The personal phone number of the study director and the experimenter including their designed substitutes were purposely put on the animal ID card in order to be contacted for emergency intervention to avoid animals suffering. To generate mice lacking TβRII in the proximal tubule, we crossed Tgfbr2[fl/fl] mice with those containing γGT-Cre[24]. To generate mice lacking TβRII in the proximal tubule and carrying the mitophagy reporter, we crossed γGT-Cre;Tgfbr2[fl/fl] mice with the mito-QC reporter mice. We used 8–12-week-old male mice.

### Reagents and antibodies

Reagent and antibody's information is listed in the Supplementary Tables 1 and 2.

### AA injury models

We intraperitoneally (ip) injected γGT-Cre;Tgfbr2[fl/fl] mice and littermate floxed controls (N10 FVB background) with 3 mg/kg aristolochic acid (AA) (Sigma-Aldrich) a total of six times over 2 weeks[24]. The mice were sacrificed 3 and 6 weeks after the last AA injection.

### Tissue staining and injury score

Kidneys were harvested, fixed in 10% formalin, paraffin or OCT embedded, and stained with hematoxylin and eosin (H&E), Sirius red or oil red O. The VECTASTAIN Elite ABC Kit (Vector Labs) was used for Pgc1α and CD3 IHC and immune-fluorescence for F4/80. Stained kidney sections were scanned using Zeiss Axio Scan. Images were quantified with ZEN software (Zeiss) and batch processing was performed using Image J (FIJI). For quantification, 10 high-power fields (HPFs) were taken per sample. Pgc1α+ nuclei in renal cortices were counted and cortical CD3+ or F4/80+ area were determined in a blinded fashion using Image J. For Sirius red and oil red O, 10 high-power fields of kidney cortices were taken per sample, and red stained area was quantified using Image J. Kidneys from mice carrying the mito-QC reporter were cryo-sectioned, and nuclei were stained with Dapi. Endogenous fluorescence was recorded using Zeiss Axio Scan. Ten HPFs were taken per sample and mito-lysosome (mCherry) was quantified with Image J. Renal injury (tubular atrophy, flattening, dilatation) and cellular infiltrates were quantified by scoring (0: no lesion; 1: 0–20% injury; 2: 20–40%; 3: 40–60%; 4: 60–80%; 5: 80–100%) in blinded manner.

### Blood urea nitrogen (BUN) measurement

At the time of euthanasia, whole blood was collected from mice, placed in heparinized tubes, and centrifuged to collect plasma that was used at the Zurich Integrative Rodent Physiology facility at the University of Zurich to determine BUN levels.

### Sample processing for spatial transcriptomics

Mice kidneys were harvested from 2 uninjured (Tgfbr2[fl/fl] and γGT-Cre;Tgfbr2[fl/fl]), 2 AA-3 weeks injured (Tgfbr2[fl/fl] and γGT-Cre;Tgfbr2[fl/fl]) and 2 AA-6 weeks injured (Tgfbr2[fl/fl] and γGT-Cre;Tgfbr2[fl/fl]) mice. The 6 samples were OCT Compound (Tissue-Tek, ref.458, SAKURA) embedded in a cryo-mold, snap frozen in liquid nitrogen using a container that has methyl butane and immediately stored at −80 °C. Thin sections of 10 μm were cut using a cryostat (LEICA CM 3050S) with the chamber set to −20 °C and the sectioning head to −15 °C, and immediately transferred on the capture area of the 10X Genomics gene expression slide. Tissues were equilibrated 1 min at 37 °C, fixed 30 min at −20 °C in pre-cooled methanol, H&E stained, and images were

captured with the slide scanner microscope (Zeiss Axio Scan). After image acquisition, tissues were permeabilized; reverse transcription and library construction were performed according to 10X Genomics instructions.

## Spatial transcriptomics analysis

After library generation using the Visium Spatial Gene Expression kit (10X Genomics) and sequencing (Functional Genomics Center Zurich, FGCZ), images and fastq files were processed and aligned using the software SpaceRanger from 10X Genomics and the mouse reference genome GRCm39. The count matrices derived from each sample were processed to filter out low-quality spots and poorly expressed genes. Normalization, scaling, and selection of highly variable genes were performed on each sample using the SCTransform method from Seurat. Principal component analysis (PCA) was performed on the merged dataset using the first 30 principal components (pcs). Clustering was performed using a community detection approach and a resolution value of 0.6. We identified positive marker genes that defined clusters compared to all other spots via differential expression using the Wilcoxon Rank Sum test. These steps were performed with the R package Seurat. Cell type's annotation was performed using the cluster markers, cell type markers from the literature, and by comparing our dataset to an external single cell reference dataset (GSE151658). Integrating spatial data with single-cell data was done by applying the 'anchor'-based integration workflow introduced in Seurat v3 that enables the probabilistic transfer of annotations from a reference to a query set. After confident annotation of our cell types, we performed trajectory inference to capture transition features in gene expression within clusters, followed by a differential expression analysis between injury time points and phenotypes for each of the cell types using the Wilcoxon test implemented in Seurat. Gene Ontology biological processes were analyzed using the R package clusterProfiler. Pathway analyses were performed using Metacore. Cell–cell communication was analyzed using the CellChat tool that is able to quantitatively infer and analyze intercellular communication networks from single-cell RNA-sequencing. Gene expression was visualized with Cloupe browser (10X Genomics). Starting from the cell counts for the different cell types, we applied Fisher's Exact test to determine whether a certain cell type is more or less abundant in the γGT-Cre;Tgfbr2$^{fl/fl}$ sample relative to Tgfbr2$^{fl/fl}$.

## snRNAseq database analysis

snRNA-Seq data were downloaded from Zenodo repository under the accession number 4059315 (Human scRNAseq on CD10+ cells). Seurat v4.1.0 in R v4.2 was used for analyses, including normalization, clustering, and differential expression. Firstly, we normalized, scaled, and centered each sample separately using *SCTransform v2*. We further applied *RunPCA, RunUMAP, FindNeighbours* with default parameters, and *FindCluster* at a resolution of 0.5. Finally, counts were recorrected with *PrepSCTFindMarkers* for differential expression. Differential gene expression was evaluated using the Wilcoxon Rank Sum test from the *FindMarkers* function. Feature plots were drawn using the *plot_density* function from the nebulosa package. Calculation of the pathway scores was performed using REACTOME genesets. We also applied Progeny algorithm to estimate the TGF-β pathway activity.

## Mitochondrial ultrastructure analysis

Kidney samples were harvested and directly immersed in 2.5% glutaraldehyde in 0.1 M sodium cacodylate buffer. Images were acquired at different magnification using a transmission electron microscope (TEM) performed on a Tecnai T12 operating at 100 keV using a side mount AMT CCD camera. Quantification of mitochondrial length and injury score (g0: no injury; g1: slight decrease of cristae number or small vacuoles; g2: severe decrease of cristae or big vacuoles formation; g3: severe decrease of cristae, big vacuoles formation and myelin

figures) was performed with Image J (9 different areas from 2 different animals per group). Unpaired Student's *t* tests was used to compare two groups. *P* values of less than 0.05 were considered statistically significant.

## Intravital mitochondria imaging

Animals were anesthetized by isoflurane (1.5–5%, in oxygen with a flow rate of 600 ml/min) and the left kidney was externalized for imaging as described previously[60]. The internal jugular vein was cannulated to allow intravenous injections of dyes and reagents. Animals were placed on a custom-built temperature-controlled stage and body temperature was monitored throughout experiments. Imaging was performed using a custom-built multiphoton microscope operating in an inverted mode, and powered by a broadband tunable laser (InSight DeepSee Dual Ultrafast Ti:Sapphire, Spectraphysics, Santa Clara, CA, USA). Intravital imaging was performed with an XLPlan N ×25/1.05 water immersion objective (Olympus, Tokyo, Japan) and emitted light was collected through four highly sensitive gallium-arsenide-phosphide photomultiplier tubes (Hamamatsu, Japan) in a non-descanned epi-fluorescence detection mode. The following excitation wavelengths were used: tetramethylrhodamine methyl ester (TMRM, 0.4 mg/kg i.v) 850 nm and Dextran Alexa 647 (2 mg/kg i.v) 1120 nm. The quantitative analysis for dextran and TMRM uptake was performed by a researcher blinded to the intervention, and each field of view (900 μm²) was captured using the same imaging setting (18 different areas from 4 different animals per group). All image processing was done in Image J (FIJI). For better visualization of the images, contrast, and brightness were modified and applied to all parts of the figures equally. Some images were post processed to improve the quality, using the FIJI gaussian blur. Data are presented as mean values (±SEM). All data were statistically evaluated using GraphPad Prism software.

## Flow cytometry

Kidneys were collected from uninjured and injured animals. After capsule removal, kidneys were immediately placed in ice-cold FACS buffer (2% FBS in PBS), mechanically dissected, and mixed with IMDM media containing collagenase IV (600 U/ml, Worthington) for 1 h at 37 °C in a shaker (150 rpm). Single cells were obtained by passing the digested kidneys through 70 μm cell strainers. Samples were washed once with FACS buffer and treated with ACK lysis buffer (150 mM NH$_4$Cl, 10 mM KHCO$_3$, and 0.1 mM Na$_2$EDTA in dH$_2$O pH 7.2) for 1 min to get rid of red blood cells. Afterward, samples were washed with FACS buffer and re-suspended in 8 ml 40% (w/v) percoll gradient solution, 70% percoll was underplayed and centrifuged for 30 min at 860 × *g* at room temperature. After centrifugation 5–6 ml percoll gradient was removed and the leukocyte ring at the inter-phase transferred to a new tube for washing with FACS buffer. Samples were divided into 3 groups for extracellular myeloid, extracellular lymphoid, and intracellular staining. Antibodies (clones, dilution, and companies) used in FACS are listed in the supplementary data. A BD FACSymphony 5 L cell cytometer was used for acquisition and FlowJo for further analysis. The yield of immune cells is always based on the step after Percoll gradient separation. Immune cells were further purified from red blood cells by a short exposure to ACK lysis buffer. The cell number, as well as the percentage of each immune cell population is defined using specific markers notably CD45. Abundance of relevant cell populations was visualized in the gating strategy, as well as in the excel sheets provided for each cell population; myeloid, lymphoid, and cytokine producing cells. Single cells were preliminary selected in terms of size with FSC/SSC gating. Singlets were further selected using SSC-H gating. To analyze cell population of interest, a Live-dead Flow Cytometry marker was used to select alive cells in our analysis. Thereafter, specific markers were used to gate and analyze for each immune cell type as stated in the gating strategy and antibody list we provided in the supplementary information.

## Mitochondria fractionation

Mitochondrial and cytosolic cell fractionation was performed using the mammalian cell mitochondria isolation kit (ThermoFisher). Briefly, proximal tubule cells were lysed in the presence of a protease inhibitor cocktail (Roche) and Dounce homogenized. Mitochondrial and cytosolic fractions were isolated by differential centrifugation and pelleted mitochondria were re-suspended in 2% CHAPS in TBS with protease inhibitors.

## Cell culture and experiments

PT cells were generated from the Immorto-mouse crossed with the Tgfbr2[fl/fl] mice[24]. PT cells were isolated from male mice and grown at 33 °C in DMEM/F12 supplemented with 2.5% fetal bovine serum, hydrocortisone, insulin, transferrin, selenium, triiodothyronine, and penicillin/streptomycin (complete PT media) with IFNγ. Prior to experiments, PT cells were moved to 37 °C and IFNγ removed to induce differentiation. Deletion of TβRII in PT was achieved by adeno-Cre treatment in vitro and verified by immunoblotting and RNAseq. PT cells were plated at 30% confluency in complete PT media. PT cells were injured with complete media containing 10–20 μM of AA for 3 to 7 days unless otherwise specified. PT cells were serum starved for 24 h prior to stimulation with TGF-β1 (R&D Systems) for up to 24 h or with 200 μM of $H_2O_2$ for 24 h.

## RNAseq

RNA isolation was performed with Machery Nagel kit including a DNase step. RNA sequencing was performed at FGCZ. Extracted RNA was prepared for sequencing using the TruSeq Stranded mRNA Library Prep assay following the manufacturer's protocol (Illumina). Sequencing was performed on the NovaSeq 6000 using the S1 Reagent Kit v1.5 (100 cycles) as per manufacturer's protocol (Illumina). Demultiplexing was performed using the Illumina bcl2fastq Conversion Software. Individual library sizes ranged from 20.7 million to 26.7 million reads. RNA sequencing analysis was performed using the SUSHI framework which encompassed the following steps: read quality was inspected using FastQC, and sequencing adapters removed using fastp; alignment of the RNA-Seq reads using the STAR aligner and with the GENCODE mouse genome build GRCm38 (patch 6, Release M23) as the reference; the counting of gene-level expression values using the 'featureCounts' function of the R package Rsubread; differential expression using the generalized linear model as implemented by the DESeq2 Bioconductor R package; Gene Ontology (GO) term pathway analysis using the hypergeometric over-representation test via the 'enrichGO' function of the clusterProfiler Bioconductor R package. All R functions were executed on R version 4.1 and Bioconductor version 3.15.

## Seahorse XFe metabolic analysis

For XFe Seahorse experiments, Mitostress test, Palmitate Oxidation Stress test, and Substrate oxidation stress tests, as well as cell density optimization and working concentration titers for each inhibitors, were performed following the manufacturer's guidelines (Agilent). For Palmitate Oxidation Stress tests, cells were incubated with substrate-limited growth media containing 0.5 μM L-carnitine overnight. The palmitate-bovine serum albumin (BSA) substrate was added right before the measurement. The injection ports included 4 μM etomoxir, 1.5 μM oligomycin, 2.5 μM FCCP, and 0.5 μM rotenone/antimycin, respectively. For Substrate Oxidation Stress assays, 30 μM BPTES and 30 μM UK5099 were used (Sigma) and all solutions were adjusted to pH 7.4 before the assay. After each experiment, cells were lysed using 1X passive lysis buffer (Promega) and Bradford assays were performed to determine protein concentrations. All experiments were analyzed using Wave software.

## ATP, lactate, and NAD+/NADH measurements

ATP, lactate, and NAD+/NADH were measured by bioluminescence assays according to the manufacturer's instructions (Promega). ATP and lactate were measured from the same cellular lysates. For each assay, optimized number of cells was seeded in 96 wells white opaque plates and incubated for 60 min in dark at room temperature before measurement. NAD+/NADH ratios were normalized to the ratio of TβRII[flox/flox] PT cells to obtain the relative ratio.

## DCFDA assay

PT cells were stained using a DCF kit at 37 °C in dark for 30 min in 1X buffer (Abcam). Afterward, TBHP (positive control) or the corresponding antioxidant (10 μM MitoQ or 1 mM of NAD+, Promega) were applied in 1X supplementary buffer at 37 °C in dark for 3 h. Cells were transferred to V-bottom 96-well plates and acquired in BD-Canto Flow Cytometer.

## qPCR

RNA from PT cells or renal cortices were isolated using the Nucleospin RNA extraction kit, following the manufacturer's instructions (Macherey Nagel). cDNA was generated using AffinityScript Multi-Temp Reverse Transcription kit (Agilent). RT-qPCR was performed using MX3000p real-time PCR machine (Agilent). Relative mRNA expression was determined by the ΔΔCT method, using Gapdh and S12 as a reference gene (after validation of a panel of housekeeping genes). Primer sequences are as follows (forward and reverse) Gapdh: 5′-AGGTCGGTGTGAACGGATTTG-3′ and 5′-TGTAGACCATGTAGTTGAGG TCA-3′; S12: 5′-GAAGCTGCCAAAGCCTTAGA-3′ and 5′-GAAGCTGC-CAAAGCCTTAGA-3′; Pgc1α: 5′-TCT CAG TAA GGG GCT GGT TG-3′ and 5′-TTC CGA TTG GTC GCT ACA CC-3′; KIM-1: 5′-CTC TAA GCG TGG TTG CCT TC-3′ and 5′-GGG CCA CTG GTA CTC ATT CT-3′; Nd1: 5′-TAG AAC GCA AAA TCT TAG GG-3′ and 5′-TGC TAG TGT GAG TGA TAG GG-3′; β-actin: 5′-GCC TTC CTT CTT GGG TAT GG-3′ and 5′-CAA TGC CTG GGT ACA TGG TG-3′.

## Immunoblotting

PT cells were lysed for protein isolation using; 25 mM Tris-HCl pH 8, 1 mM EDTA, 150 mM NaCl, 1% NP-40, 1 mM $Na_3VO_4$, and protease inhibitor cocktails (Sigma). The tissue lysis buffer additionally contained 1 mM PMSF, 1 μg/ml aprotinin, 1 μg/ml leupeptin, and 1 μg/ml pepstatin A. Proteins of both tissue and cell lysates were separated by SDS-PAGE and transferred to nitrocellulose membranes. All primary antibodies were dissolved in 5% BSA solution and incubated with membranes overnight at 4 °C. The following primary antibodies were used: Pink1 (Novus), LC3A (CST), Pgc1α (Merck), cleaved caspase 9 (CST), phospho-Smad3 (CST), total OXPHOS (ab110413, Abcam), Polγ (Santa Cruz), Tom20 (Santa Cruz), α-tubulin (GT114, GeneTex), polyclonal goat anti-rabbit-HRP (31460, Pierce), polyclonal goat anti-mouse-HRP (31430, Pierce). Following addition of HRP substrate (NEL113001EA, PerkinElmer), chemiluminescence was recorded with a LAS-400 mini Luminescent Image Analyzer and Image J was used for quantification.

## Statistical analysis

If not stated, unpaired Student's $t$ tests was used to compare two sets of data, and two-way ANOVA followed by Sidak's multiple comparisons was used to compare 4 groups (FACS), with *$p < 0.05$ considered significant. At least 3 independent experiments or biological replicates are performed for each in vitro experiments unless otherwise stated. Results are presented as mean ± SEM.

## Reporting summary

Further information on research design is available in the Nature Portfolio Reporting Summary linked to this article.

## Data availability

RNAseq raw and metadata are available at both GEO accession GSE225545 and Zenodo repository (https://zenodo.org/record/7912879#.ZGDp5oRByUl). Visium data and analysis are available at Zenodo repository (https://doi.org/10.5281/zenodo.7635958). Online scRNAseq data used for cell type annotation in Visium were from GSE151658[32]. snRNA-Seq data were from Zenodo repository under the accession number 4059315[48]. All other relevant data supporting the key funding of this study are available within the article and its Supplementary information files. Source data are provided with this paper.

## Code availability

All analysis were performed based on the Sushi uzh/sushi: SUSHI: Supporting User for SHell script Integration (github.com) and ezRun uzh/ezRun: an R meta-package for the analysis of Next Generation Sequencing data (github.com)[61].

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

## Acknowledgements

This work was supported by the Swiss National Science Foundation (AMBIZIONE grant PZOOP3_179916 and the Swiss Federal Government Excellence Scholarship; NCCR "Kidney.CH" junior grant to S.N.K.); the Swiss National Science Foundation (310030_184813 and NCCR "Kidney.ch", 183774, to R.H.W.); the Peter Hans Hofschneider Professorship for Molecular Medicine and the Swiss National Science Foundation (PCEGP3_194216 to C.S.); the NIH grant R01-DK-108968. We thank Patrick Spielmann for the technical support and Dr. Ian G. Gantley for the MitoQC mice.

## Author contributions

M.K., J.V., M.B., Y.M., J.G., and S.N.K. performed the in vivo and in vitro experiments and data analysis. D.G.R., H.R., P.J.L., and S.N.K. designed and performed spatial and bulk-RNAseq data analysis. C.S. was responsible for immune cell profiling analysis by FACS. A.H. was responsible for intravital analysis of mitochondria function by multiphoton microscopy. D.L. performed the analysis of online database of human healthy and CKD patients. C.A.A. contributed spatial transcriptomics experiment design. L.G. provided the conditional knockout mice and discussed the results. R.H.W. discussed the results. S.N.K. supervised the project and wrote the manuscript with input from all authors.

## Competing interests

The authors declare no competing interests.
