## [Peer review file · Nature Communications]

REVIEWER COMMENTS

Reviewer #1 (Remarks to the Author):

In this manuscript, the authors show how TGF- β signaling affects mitochondrial quality control mechanisms, fibrosis, fatty acid metabolism, and inflammation in the proximal tubules and how these alterations attenuate mitochondrial dysfunction and other markers of chronic kidney disease. Overall, this is a very good and important paper.

However, the manuscript is too dense and tries to cover too many issues. As such, the manuscript loses focus and data integrity. Finally, the Discussion overstates the stated relationships and conclusions.

Below are some are some deficiencies and questions:

- Figure one legend includes the statement “dots represent the number of animals per group”. However, not all figures provide an explanation about the number of animals per group or the meaning of dots.
- Figure 1j, BUN levels seem not to be different statistically.
- Figures 3F, 3G, 3M, and 3N was not subjected to statistical analysis nor the conclusion
- Figure 3b, densitometry analysis of ndufb8, sdhb, and mtco1 are shown. However, only ndufb8 is statistically reduced gGTCre;Tgfbr2fl/fl compared to Tgfbr2fl/fl. Please show densitometry of all complexes or only shown densitometry of the ones statistically different.
- In figure 3c, only 4 complexes are shown in the immunoblot. If this immunoblot was performed still using antibody # ab110413, all 5 complexes should be shown. Is there a reason why the complex III band is not shown?
- Evidence support GAPDH is an unstable housekeeping gene in transcript level in ischemia and other kidney disease (<https://doi.org/10.1371/journal.pone.0233109>). As mentioned, PGC1 α negatively regulates TGF β and is the main transcriptional regulator of mitochondrial biogenesis. It would be interesting (Fig-4a) to assess the mRNA levels using β -actin as a house keeping gene.
- TGF- β , Transforming growth factor beta (TGF- β) is not spelled anywhere in the manuscript.
- Figure 1f, KIM-1 is measured by mRNA. It would be important to see protein levels of KIM-1.
- Figure 2a, the meaning of symbols “N”, “BB”, or the red arrows are not explained on the figure legend or anywhere in the manuscript.
- Line 208, authors mention mRNA and protein studies for PGC-1 α , but did not reference “extended data figure 6”.
- Figure 4f-h, it would be an interesting addition to show the rate of conversion of LC3II:I. An increase of the LC3II:I ratio and has been identified as a consistent marker of impaired autophagy. By looking at the

immunoblot, it is clear that the LC3II:I ratio is increasing in the γ GT-Cre; Tgfr2 fl/fl group compared to Tgfr2 fl/fl.

(2021) Guidelines for the use and interpretation of assays for monitoring autophagy (4th edition)¹, *Autophagy*, 17:1, 1-382, DOI: 10.1080/15548627.2020.1797280

- Line 216 states that mitochondrial markers such as tfam were increased in T β R1I^{-/-} PT cells compared to T β R1Iflox/flox PT cells, but data is not shown. Showing markers such as tfam would strengthen the argument
- Methods reporting is not sufficient and is often missing information. Some reagents/chemicals include a catalog number such as in line 558 “total OXPHOS (ab110413, abcam)”. However, a large number of chemicals, reagents or kits do not include a catalog number or origin.

Reviewer #2 (Remarks to the Author):

This manuscript by Kayhan et. al seeks to uncover how TGF-B signaling affects PT mitochondria function in CKD. The authors choose to tackle a well-worn topic, but do so with a tour de force approach, comprehensively defining the metabolic perturbations, transcriptomic alterations, and injury phenotypes of TGFBR1I knockout in an aristilochic acid animal model. The authors perform additional validation in a proximal tubule cell culture model and draw upon existing clinical datasets to provide further corroboration of their results. This work builds upon their prior investigations in a 2017 JASN publication using the same murine model.

The authors conduct a bevy of experiments and analyses: 1) histology and BUN, 2) immunofluorescence of immune cell infiltration and Pgc1a, 3) spatial transcriptomics, 4) use publicly available sc or sn RNAseq of murine and human kidneys including cell chat analyses, 5) electron microscopy of mitochondria, 6) FACS for immune cell infiltration, 7) intravital imaging for dextran uptake, 8) Seahorse cell culture, 9) RNAseq of a PT cell culture model with TGFBR1I knock out, 10) metabolic bioluminescence assays, 11) DCFDA cell assay, 12) RT-PCR and immunoblotting of key genes and proteins.

The strengths of this investigation include strong orthogonal validation of their murine model with a vast array of techniques. The effect of TGFBR2 KO in a CKD model is comprehensively investigated, mixing recent technologies like spatial transcriptomics with classic ones. The potential of this manuscript has been nearly maximized.

The main critique is the novelty of the question addressed. Others have examined mitochondrial dysfunction associated with TGF-beta in kidney disease. In fact, this topic has been the subject of multiple reviews, including a 2017 review in *Kidney International* (PMID: 28890325) and a 2012 review in *Seminars in Nephrology* (PMID: 22835461). However, most of the prior investigations lack the comprehensive approach of the work by Kayhan et al or do not directly focus on the TGFBR2. One investigation in *Disease models and mechanisms* (PMID: 34431499) performs similar work investigating TGFBR3 and another in *PLoS ONE* assesses the effects of TGFBR1 inhibition (31318905). The work by Kayhan et al is more comprehensive than either.

Specific critiques:

1. The abstract is general and fails to convey the experimentation conducted or results obtained. Humans and patients are mentioned in the abstract, but not mice, yet mouse and cell culture models constitute the bulk of the results.
2. Figure 1 – the advantages of ST are not maximally leveraged in figure 1. The injury markers of Fig 1D appear broadly upregulated. Figure 1B/C assigns a single cell type to each spot. There are deconvolution methods which would facilitate colocalization analyses of the immune cell infiltrate with the various epithelial cell types.
3. Figure 2 D/E – are these S3 PT's depicted?
4. Figure 3A – how do you define PT specific spots? It would be helpful to add a couple feature plots of genes in these pathways that are detected in PT spots.
5. Fig 5A – did Th1 inflammatory cells colocalize transcriptomically with PT S3 signature?
6. Fig 6B – the relevance of this is not apparent, please provide greater interpretation of these clusters.
7. Methods – please provide the number of ST samples and spots. Pre-sequencing and Sequencing QC metrics should be provided.

Reviewer #3 (Remarks to the Author):

In the manuscript titled “Intrinsic TGF- β signaling attenuates proximal tubule mitochondrial injury and inflammation in chronic kidney disease”, the authors built on their earlier work on TGF receptor signaling. Here, they investigated the interplay of aristolochic acid-mediated injury of the kidney and the targeted deletion of the Tgfbr2 receptor in the proximal tubules in mice now focusing on the consequences of Tgfbr2 receptor deletion on mitochondrial function and tissue inflammation status. In a second line of experiments, the authors complemented their studies in mice by investigating conditionally immortalized proximal tubule cells that lack the T β RII receptor. In a third line of

experiments, the authors chose a spatial transcriptomics approach to identify cells in kidney sections by their transcript profiles and with that to gather spatiotemporal information on cellular and transcriptional changes following the injury by aristolochic acid (AA) in the *Tgfr2* receptor deletion mouse.

General comments:

This study updates our current understanding of the adaptability of TGF- β signaling by shedding light on its role in mitochondrial homeostasis and renal immunity not only in the context of the presented kidney injury model but also by pointing to similarities to the human CKD phenotype. Hence, due to the high prevalence of CKD in our society and the importance of TGF- β signaling in this context, the manuscript will be of great interest to the clinical and research community.

The biochemical work involving conditionally immortalized proximal tubule cells is thoroughly and carefully carried out, documented, and presented. The same is the case for all histology and imaging experiments on mouse samples.

In contrast, the transcriptomic experiments fall somewhat short. The reviewer appreciates that the authors highlight shortcomings in this area in the discussion section of their manuscript. Particularly, the use of single cell RNA sequencing (scRNAseq) data that stem from a biological experiment unrelated to the one pursued here are a concern as they were used to identify cell types in the spatial transcriptomics data, which formed the base of subsequent analyses and experimental steps. However, the observations made by the authors were substantiated by independent approaches they had carried out to test their hypotheses. Still, the chosen approach raises concerns. It would have been ideal if the authors could have generated scRNAseq data of their mouse model and used that as a source of single cell information. However, it is understood that such experiments are very expensive and hence not easily accessible. Spatial transcriptomics is an up-and-coming technology that has a lot to offer, and the authors made the best of it in terms of what was available to them.

The transcriptomic experiments fall short for another reason. The manuscript provides only very sparse information on sample generation, source of data (no specifics to which of the data from GEO/Zenodo were used), and processing (how were the different datasets processed? How many cells went into the analysis? What percentage of doublets/mito/ribo transcripts were excluded? etc.). For example, it is not clear to the reviewer whether frozen or fixed samples were used for the Visium approach, what section thickness was chosen, how sections were selected etc. This technology is young and even less accessible to many laboratories. Hence, it is essential that details are provided at a level that is sufficient to allow the reader to understand the experimental design and to reproduce the data processing steps. With regards to reproducibility, it is now common practice to provide key steps of data processing and visualization on open-access platforms such as Github.

Minor comments:

Title: The authors might want to limit the statement to the model they investigated i.e., AA mouse model of CDK. Alternatively, the authors may want to mention the use of a mouse model in the Abstract.

The reviewer felt at times frustrated about the frequent reference to 'data not shown' in the manuscript.

A bit more background information would be helpful to the non-initiated reader. For example, the AA model and the effects of AA on the kidney are not introduced. Not everybody is aware of the mito-QC mouse. What does it offer? Inconsistency when describing molecular players i.e., Pink1 is described but not LC3A-I.

There is a typo in line 275: "CD8+ cells were augmented in gGT-Cre; Tgfbr2fl/fl kidneys". It should read: "... augmented in Tgfbr2fl/fl kidneys."

Line 387: It should say: The mice were euthanized instead of 'sacrificed'

Methods:

- It is not clear what the background of the Tgfbr2fl/fl mice is. Does it match that of the mice from which the scRNAseq data were borrowed?
- Was proteinuria followed?
- At what voltage were the electron micrographs taken? What system, detector?

Figures:

- Figure 1 and ED (Extended Data) Figure 1:

It is not clear what the number of animals was in the experiments that lead to panels 1b and 1c i.e., how variable are the proportions of cell types in the tested groups, and, as the study focuses on male mice, how representative are the findings in general?

- Fig 1a: at the resolution provided, the text overlaid onto the UMAP is not legible and, in the case of Myo/St.mixed, seems not to be in the correct location

- Fig 1b: The text overlaid on the spatial feature plot are not legible; the colors for S3 and Endo/Glom are hard to distinguish, especially with the H&E background. It is not clear which time point after aristolochic acid treatment is reflected here.

- Fig 1c: colors do not match color scheme in panels a and b

- Fig 1d: the resolution of the H&E and panels below is rather low; the Loupe browser images show horizontal bands with distortions

- Fig 1e: resolution is marginal

- ED Fig 1a, b: which genes shown in panel b correspond to which violin plot in panel a? Panels a and b shows 16 clusters. To what areas in the UMAP shown in Fig1a do they map to i.e., could the authors add a panel in the ED Fig1 that would show the location of the 16 clusters in an UMAP plot?

- Fig 1i, j: in both panels, for the Cre/fl.fl Tgfb β 2 mice, there are two outliers. Do the data come from the same mice?

• Figure 2 and ED (Extended Data) Figure 2:

- ED Fig 2b: color scheme makes it hard to identify data points that associate with the term 'Absent'

- Fig 2a: the resolution of the EM images is marginal; the text overlaid on the EM micrographs is not explained in the figure legend; scale-bar dimension for the higher-magnification images is not provided

- Fig 2f: how do the data points relate to number of areas per animals per group?

- Fig 2g: image resolution is marginal

• Figure 4 and ED (Extended Data) Figure 4:

- ED Fig4: panel d, the purpose of the GFP channel in imaging is not immediately clear from the text or figure legend

• Figure 5 and ED (Extended Data) Figure 5:

- ED Fig5: typos in headers; the reviewer does not find the cartoon-like Metacore representation very helpful. In panel a, the 'meter' that shows the expression levels obscures some of the components it associates with; a table format would be easier to follow; however, the depiction of the pathways is appreciated.

• Figure 6 and ED (Extended Data) Figure 6:

- ED Fig6: the image of the western blot is suggesting that the antibody is detecting two bands for PGc1-alpha in T β RIIflox/flox; is that typical?

- Fig 6e: typo in header of first panel

- Figure 7 and ED (Extended Data) Figure 7:

- ED Fig 7: panel a, the resolution of the heat map is marginal, the two groups at 6 weeks are hard to discern by choice of color; I think there are better plotting options to show gene expression along trajectory (?). Panel b, this panel needs more explanation, please add description in text/figure legend; are there data for the 6-week time point? If so, how does it compare with the earlier time point? Panel c needs more information; to a non-initiated it is not clear what LTL is referring to.

In response to the reviewers' comments, the following changes have been made:

Reviewer #1

In this manuscript, the authors show how TGF- β signaling affects mitochondrial quality control mechanisms, fibrosis, fatty acid metabolism, and inflammation in the proximal tubules and how these alterations attenuate mitochondrial dysfunction and other markers of chronic kidney disease. Overall, this is a very good and important paper. However, the manuscript is too dense and tries to cover too many issues. As such, the manuscript loses focus and data integrity. Finally, the Discussion overstates the stated relationships and conclusions.

Response: We appreciate the reviewer's comment and constructive critiques. We made efforts to contract the text and improve the stated relationships and conclusions.

Below are some are some deficiencies and questions:

1) Figure one legend includes the statement "dots represent the number of animals per group". However, not all figures provide an explanation about the number of animals per group or the meaning of dots

Response: We thank the reviewer for this remark and apologize for omitting information on the meaning of dots and animal number in other figures. Dots and squares in the plots represent the number of animals per group or of *in vitro* experiment replicates in all figures unless otherwise stated. We provided this information in all figure legends.

2) Figure 1j, BUN levels seem not to be different statistically

Response: We thank the reviewer for the comments. The quantification was statistically significant regardless data distribution. However, we added 4 Tgfb2^{fl/fl} and 3 γ GTCre;Tgfb2^{fl/fl} mice to improve the statistical significance.

3) Figures 3F, 3G, 3M, and 3N was not subjected to statistical analysis nor the conclusion

Response: We thank the reviewer for pointing this remark out, and we understood that the reviewer actually referred to the Seahorse graphs in the "Extended Data Fig. 3" which has become **Extended Data Fig. 4** in the revised version of the manuscript. The panels (f and g) are representative graphs of oxygen consumption rate (f) and extracellular acidification (g). The quantifications of relevant metabolic parameters are in the following panels (h-l). We

added quantification of relevant metabolic parameters for panels (m and n) in the Supplementary Fig. 3 and 4.

4) Figure 3b, densitometry analysis of *ndufb8*, *sdhb*, and *mtco1* are shown. However, only *ndufb8* is statistically reduced *gGTCre;Tgfb2fl/fl* compared to *Tgfb2fl/fl*. Please show densitometry of all complexes or only shown densitometry of the ones statistically different

Response: We thank the reviewer for this comment. We included densitometry of *sdhb* and *mtco1* to show their reduction trends in the γ GT-Cre;*Tgfb2^{fl/fl}* mice compared to the floxed mice. However, we agree with the reviewer and removed densitometry analysis of other subunits in Figure 3.

5) In figure 3c, only 4 complexes are shown in the immunoblot. If this immunoblot was performed still using antibody # ab110413, all 5 complexes should be shown. Is there a reason why the complex III band is not shown?

Response: We thank the reviewer for this important remark and apologize for wrongly labelling complex III as complex IV in the panel c. We corrected ETC subunit labelling in panel c. Regarding complex IV subunit (MTCO1), its expression is weak in our PT cells (faint bands). Moreover, the antibody manufacturer mentioned in the datasheet (*) that “the MTCO1 is highly hydrophobic. Its predicted MW is 57kDa but it differently migrates in different gel systems (<https://www.abcam.com/total-oxphos-rodent-wb-antibody-cocktail-ab110413.html>). However, MTCO1 bands with an apparent MW of approx. 35 kDa are visible after long exposure at higher exposure followed by image adjustment (red arrows). Please find below an example of gels where red arrows indicate faint MTCO1 bands.

6) Evidence support GAPDH is an unstable housekeeping gene in transcript level in ischemia and other kidney disease (<https://doi.org/10.1371/journal.pone.0233109>). As mentioned, PGC1 α negatively regulates TGF β and is the main transcriptional regulator of mitochondrial biogenesis. It would be interesting (Fig-4a) to assess the mRNA levels using β -actin as a housekeeping gene.

Response: We appreciate the reviewer's comment and we understand that, even though its mRNA expression was stable in our experiments, using *Gapdh* as housekeeping gene in the context of cell metabolism can be misleading. We therefore provided additional experiments and tested the ribosomal protein Rps12 (S12) and β -actin (*Actb*) as housekeeping genes. Based on these results and a previous study (Dahl SL et al. *Acta Physiol.*, 2022 234(3):e13768), we replaced *Gapdh* by S12/*Actb* as housekeeping genes in the qPCR analyses. Please find below, the comparison between these 3 selected housekeeping genes.

7) TGF- β , Transforming growth factor beta (TGF- β) is not spelled anywhere in the manuscript.

Response: We thank the reviewer for this point. We spelled TGF- β at the first mention in the introduction.

8) Figure 1f, KIM-1 is measured by mRNA. It would be important to see protein levels of KIM-1.

Response: We thank the reviewer for this comment. We provided KIM-1 immunofluorescence staining and quantification in the Extended Data Fig. 2.

9) Figure 2a, the meaning of symbols “N”, “BB”, or the red arrows are not explained on the figure legend or anywhere in the manuscript.

Response: We apologize for these omissions. We added the meaning and explanation of these symbols in Figure 2's legend.

10) Line 208, authors mention mRNA and protein studies for PGC-1 α , but did not reference “extended data figure 6”.

Response: We thank the reviewer for this remark. We referenced Extended Data Fig.8a, b in line 212. For better legibility, we also referenced Extended Data Fig.8 in line 208 as suggested by the reviewer.

11) Figure 4f-h, it would be an interesting addition to show the rate of conversion of LC3II:I. An increase of the LC3II:I ratio and has been identified as a consistent marker of impaired autophagy. By looking at the immunoblot, it is clear that the LC3II:I ratio is increasing in the γ GT-Cre; Tgfbr2 fl/fl group compared to Tgfbr2 fl/fl. (2021) Guidelines for the use and interpretation of assays for monitoring autophagy (4th edition)1, Autophagy, 17:1, 1-382, DOI: 10.1080/15548627.2020.1797280

Response: We thank the reviewer for raising this important point. The LC3A-II:LC3A-I ratio tends to increase in γ GT-Cre;Tgfbr2^{fl/fl} group compared to Tgfbr2^{fl/fl} as rightly noticed by the reviewer. In Figure 4f-h, we wanted to emphasize that γ GT-Cre;Tgfbr2^{fl/fl} have decreased LC3A-I compared to Tgfbr2^{fl/fl} mice, whereas LC3A-II conversion was not significantly different between genotypes, suggesting a decrease in whole LC3A expression in γ GT-Cre;Tgfbr2^{fl/fl} mice. As mentioned in the 2021 Guidelines for the use and interpretation of autophagy, Atg8/LC3A-I expression and conversion to Atg8-PE/LC3A-II can be affected by multiple

parameters including the speed of LC3A-I/LC3-II conversion and of LC3A-II lysosomal degradation. The optimal way to evaluate autophagy using LC3A would necessitate the use of specific controls, notably pharmacological inducers of autophagy (as positive control), as well as proteasomal and lysosomal degradation inhibitors, which is out of the scope of this study. However, assessment of mitophagy using mito-QC reporter mice (Figure 4d) would suggest a more efficient lysosomal degradation axis in *Tgfbr2^{fl/fl}* and may explain the lower proportion of LC3A-II in the *Tgfbr2^{fl/fl}* despite higher expression of LC3A-I.

12) Line 216 states that mitochondrial markers such as *tfam* were increased in *TβRII^{-/-}* PT cells compared to *TβRIIflox/flox* PT cells, but data is not shown. Showing markers such as *tfam* would strengthen the argument

Response: We thank the reviewer for this comment. We added *Tfam* transcript expression data in the Extended Data Fig. 8.

13) Methods reporting is not sufficient and is often missing information. Some reagents/chemicals include a catalog number such as in line 558 “total OXPHOS (ab110413, abcam)”. However, a large number of chemicals, reagents or kits do not include a catalog number or origin

Response: We apologize for these omissions. We added supplementary information regarding all reagents and antibodies information in the Supplementary tables 1 and 2.

Reviewer #2

This manuscript by Kayhan et. al seeks to uncover how TGF-B signaling affects PT mitochondria function in CKD. The authors choose to tackle a well-worn topic, but do so with a tour de force approach, comprehensively defining the metabolic perturbations, transcriptomic alterations, and injury phenotypes of *TGFBR2* knockout in an aristolochic acid animal model. The authors perform additional validation in a proximal tubule cell culture model and draw upon existing clinical datasets to provide further corroboration of their results. This work builds upon their prior investigations in a 2017 JASN publication using the same murine model. The authors conduct a bevy of experiments and analyses: 1) histology and BUN, 2) immunofluorescence of immune cell infiltration and *Pgc1a*, 3) spatial transcriptomics, 4) use publicly available sc or sn RNAseq of murine and human kidneys including cell chat analyses,

5) electron microscopy of mitochondria, 6) FACS for immune cell infiltration, 7) intravital imaging for dextran uptake, 8) Seahorse cell culture, 9) RNAseq of a PT cell culture model with TGFBR2 knock out, 10) metabolic bioluminescence assays, 11) DCFDA cell assay, 12) RT-PCR and immunoblotting of key genes and proteins. The strengths of this investigation include strong orthogonal validation of their murine model with a vast array of techniques. The effect of TGFBR2 KO in a CKD model is comprehensively investigated, mixing recent technologies like spatial transcriptomics with classic ones. The potential of this manuscript has been nearly maximized. The main critique is the novelty of the question addressed. Others have examined mitochondrial dysfunction associated with TGF-beta in kidney disease. In fact, this topic has been the subject of multiple reviews, including a 2017 review in *Kidney International* (PMID: 28890325) and a 2012 review in *Seminars in Nephrology* (PMID: 22835461). However, most of the prior investigations lack the comprehensive approach of the work by Kayhan et. al or do not directly focus on the TGFBR2. One investigation in *Disease models and mechanisms* (PMID: 34431499) performs similar work investigating TGFB3 and another in *plos one* assesses the effects of TGFBR1 inhibition (31318905). The work by Kayhan et al is more comprehensive than either.

Response: We appreciate the reviewer's comment and constructive critiques. We agree that several studies speculate on the role of systemic TGF- β signaling on mitochondria and inflammatory response; however, our study is the first study that demonstrates with strong orthogonal validation how TGF- β signaling affects the proximal tubule mitochondrial homeostasis and inflammatory in CKD. Our study reveals the potential role of S3T2 PT cells in post-injury inflammatory response. Moreover, our online database analysis suggests a possible role of intact PT TGF- β signaling to ensure adaptive response to CKD in humans.

Specific critiques:

1) The abstract is general and fails to convey the experimentation conducted or results obtained. Humans and patients are mentioned in the abstract, but not mice, yet mouse and cell culture models constitute the bulk of the results.

Response: We thank the reviewer for the remark and agree that the abstract failed to provide details on the experiments conducted and results. We rewrote the abstract accordingly.

2) Figure 1 – the advantages of ST are not maximally leveraged in figure 1. The injury markers of Fig 1D appear broadly upregulated. Figure 1B/C assigns a single cell type to each spot.

There are deconvolution methods which would facilitate colocalization analyses of the immune cell infiltrate with the various epithelial cell types.

Response: We thank the reviewer for this important point. The Cloupe browser images in Fig. 1d provide an overall signal/intensity pattern of injury and fibrosis markers and validate the use of this technique to assess injury induction. These injury and fibrosis signals are confirmed in Fig.1f and Extended Data Figure 2 for Kim-1 (Havcr1 mRNA and immunofluorescence staining) and Fig.1h for collagen accumulation (Sirius red staining). We agree that a deconvolution could be run with the expression data of these individual cell types (single cell or sorted bulk cells). We chose integration over deconvolution methods in our spatial transcriptomics since the former are specifically designed to be robust to different noise models that characterize spatial and single-cell datasets, and have superior performance. Therefore, each spot derived from the unsupervised clustering, the expression of marker genes described in the literature and the integration with the external single cell reference dataset GSE151658 using the 'anchor'-based integration workflow introduced in Seurat v3 ([https://www.cell.com/cell/fulltext/S0092-8674\(19\)30559-8](https://www.cell.com/cell/fulltext/S0092-8674(19)30559-8)) that enables the probabilistic transfer of annotations from a reference to a query set. The output for each spot is a probabilistic classification for each of the scRNA-seq derived cell types, meaning that the reported celltype represents the dominant cell type at each spot.

3) Figure 2 D/E – are these S3 PT's depicted?

Response: We thank the reviewer for raising this important point. The live microscopy does not reach deep into the tissue (including outer medulla), so the S3 PT cells, especially those dwelling in the outer medulla, are not depicted. However, the proximal tubule is divided in two parts: the convoluted tubules (S1 and part of S2) dwelling in the cortex and the parse recta (part of S2 and S3) located at the cortico-medullary junction and in the outer medulla. i.e. S2 is the transition between S1 and S3. Recently, Polesel M et al., Nat Commun. 2022 (<https://doi.org/10.1038/s41467-022-33469-5>) revealed that the Ggt1 (gene) promoter driving Cre-expression in our mice is also active in the S2 segment, suggesting that not only S3 PT cells are concerned by Tgfbr2 deletion, but also part of S2 PT cells. Thus, the Figure 2 D/E depicts S1 and Ggt1 expressing proximal tubules including the transitional S2. Although Tgfbr2 is deleted in the cortico-medullary junction (part of S2 and S3 segment), showing the whole cortical proximal tubule adds a pathophysiological relevance to our findings, since increased vulnerability in the S3 segment (in the absence of Tgfbr2) can worsen S1 and S2 injury.

4) Figure 3A – how do you define PT specific spots? It would be helpful to add a couple feature plots of genes in these pathways there are detected in PT spots.

Response: We thank the reviewer for this point. We included all cell types in Fig. 3A. PT specific feature plots are shown in Extended Figure 1. Please find below the dot plots that are included in the Extended Figure 1.

5) Fig 5A – did Th1 inflammatory cells colocalize transcriptomically with PT S3 signature?

Response: We thank the reviewer for raising this point. The S3 cluster (in the deep cortex) disappears under injury as we expected. The remaining part, the S3T2, dwells in the outer medulla and is involved in immune cell activation under injury. Based on cluster resolution on kidney slides, we can also observe the proximity between the S3T2 and clusters containing lymphocytes (Myo/St. mixed). However, one of the obvious limitations of spatial transcriptomics (Visium) is its low resolution (spot diameter is 55µm). Although we can observe dot proximity, it is not proof for single cell proximity. However, we observed physical proximity of CD3+ positive T cells and PT cells in the cortico-medullary junction. Please refer to the image below.

6) Fig 6B – the relevance of this is not apparent, please provide greater interpretation of these clusters

Response: We thank the reviewer for this comment. We replaced Fig. 6b by a plot providing the percentage of PT cells per clusters in healthy and CKD biopsies. No significant difference in the overall PT cell proportion was found between healthy and clusters.

7) Methods – please provide the number of ST samples and spots. Pre-sequencing and Sequencing QC metrics should be provided

Response: We thank the reviewer for this important remark and apologize for these omissions. We added a supplementary table 3 with the requested information.

Reviewer #3

General comments:

§ This study updates our current understanding of the adaptability of TGF- β signaling by shedding light on its role in mitochondrial homeostasis and renal immunity not only in the context of the presented kidney injury model but also by pointing to similarities to the human CKD phenotype. Hence, due to the high prevalence of CKD in our society and the importance of TGF- β signaling in this context, the manuscript will be of great interest to the clinical and research community.

Response: We appreciate the reviewer's comment.

§ The biochemical work involving conditionally immortalized proximal tubule cells is thoroughly and carefully carried out, documented, and presented. The same is the case for all histology and imaging experiments on mouse samples.

Response: We appreciate the reviewer's comment.

§ In contrast, the transcriptomic experiments fall somewhat short. The reviewer appreciates that the authors highlight shortcomings in this area in the discussion section of their manuscript. Particularly, the use of single cell RNA sequencing (scRNAseq) data that stem from a biological experiment unrelated to the one pursued here are a concern as they were used to identify cell types in the spatial transcriptomics data, which formed the base of subsequent analyses and experimental steps. However, the observations made by the authors were substantiated by independent approaches they had carried out to test their hypotheses. Still, the chosen approach raises concerns. It would have been ideal if the authors could have generated scRNAseq data of their mouse model and used that as a source of single cell information. However, it is understood that such experiments are very expensive and hence not easily accessible. Spatial transcriptomics is an up-and-coming technology that has a lot to offer, and the authors made the best of it in terms of what was available to them.

Response: We thank the reviewer for highlighting the limitations of the spatial transcriptomics that we did emphasize in the discussion section. We also appreciate that the reviewer understood that such experiments are very expensive and not easily accessible. However, we invested considerable effort to get best of it with regard to our assigned goals. While the spatial transcriptomics analyses could have been presented in a separate manuscript, we used it as a tool to point out the spatio-temporal insights in proximal tubule cells and immune cells interactions in late acute to chronic injury transition.

§ The transcriptomic experiments fall short for another reason. The manuscript provides only very sparse information on sample generation, source of data (no specifics to which of the data from GEO/Zenodo were used), and processing (how were the different datasets processed? How many cells went into the analysis? What percentage of doublets/mito/ribo transcripts were excluded? etc.). For example, it is not clear to the reviewer whether frozen or fixed samples were used for the Visium approach, what section thickness was chosen, how sections were selected etc. This technology is young and even less accessible to many laboratories. Hence, it is essential that details are provided at a level that is sufficient to allow

the reader to understand the experimental design and to reproduce the data processing steps. With regards to reproducibility, it is now common practice to provide key steps of data processing and visualization on open-access platforms such as Github.

Response: We thank the reviewer for this important point. Regarding sample processing for the spatial transcriptomics experiments, we provided the pre-sequencing (sample processing) and sequencing information requested by the reviewer in the supplementary table 3 and in the online methods section “*Samples processing for Spatial Transcriptomics*”. Regarding the human data sources, we provided additional information of the used dataset (accession number of the Zenodo repository is 4059315, Human scRNAseq on CD10+ cells, epithelial cells only, PMID 33176333) in the online method “*snRNAseq Database Analysis*”. For snRNAseq data, Seurat v4.1.0 in R v4.2 was used for analyses, including normalization, clustering and differential expression. Firstly, we normalized, scaled and centered each sample separately using *SCTransform v2*. We further applied *RunPCA*, *RunUMAP*, *FindNeighbours* with default parameters, and *FindCluster* at a resolution of 0.5. Finally, counts were recorrected with *PrepSCTFindMarkers* for differential expression. Differential gene expression was evaluated using the Wilcoxon Rank Sum test from the *FindMarkers* function. Feature plots were drawn using the *plot_density* function from the *nebulosa* package. Calculation of the pathway scores was performed as previously described using REACTOME genesets. In this information is present in the online methods.

Minor comments:

1) Title: The authors might want to limit the statement to the model they investigated i.e., AA mouse model of CDK. Alternatively, the authors may want to mention the use of a mouse model in the Abstract.

Response: We agree with the reviewer comment and mentioned the used injury model in the abstract.

2) The reviewer felt at times frustrated about the frequent reference to ‘data not shown’ in the manuscript.

Response: We thank the reviewer for this comment and we apologize for this inconvenience. We removed all “data not shown” and provided these data in the appropriate Extended Data

Figures (2, 7 and 8) and Supplementary Figures (4, 5 and 8), with the exception of the glycolysis markers to keep the manuscript's focus.

3) A bit more background information would be helpful to the non-initiated reader. For example, the AA model and the effects of AA on the kidney are not introduced. Not everybody is aware of the mito-QC mouse. What does it offer? Inconsistency when describing molecular players i.e., Pink1 is described but not LC3A-I.

Response: We apologize for these omissions. We provided additional information on the aristolochic acid model and described the mito-QC reporter mouse and defined LC3A.

4) There is a typo in line 275: "CD8+ cells were augmented in γ GT-Cre; Tgfbr2^{fl/fl} kidneys". It should read: "... augmented in Tgfbr2^{fl/fl} kidneys."

Response: We thank the reviewer for this comment. "CD8+ cells were augmented in γ GT-Cre; Tgfbr2^{fl/fl} kidneys" but not statistically significant based on the whole quantification. We agree that the proportion of CD8 cells in the lymphocyte population, depicted in the FACS graph, suggested a higher proportion of Cd8+ cells in Tgfbr2^{fl/fl}. However, the proportion of pro-inflammatory (TNFa+ and IFNg+) CD8 cells is lower in Tgfbr2^{fl/fl} compared to γ GT-Cre; Tgfbr2^{fl/fl}.

5) Line 387: It should say: The mice were euthanized instead of 'sacrificed'

Response: We appreciate the reviewer's comment. We replaced sacrificed by euthanized.

Methods:

6) It is not clear what the background of the Tgfbr2^{fl/fl} mice is. Does it match that of the mice from which the scRNAseq data were borrowed?

Response: We thank the reviewer for raising this important point. The mice in the scRNAseq data were in the C57BL/6 background while our mice are in the FVB background. The scRNAseq data used a sepsis model of kidney injury. Indeed, LPS injection has the advantage over other sepsis models to recapitulate human sepsis. Importantly, this model does not show age and strain variability in mice (<https://doi.org/10.1172/JCI39421> and <https://doi.org/10.1016/j.kint.2019.12.024>). Moreover, our previous data on fibrosis induction revealed similar pathophysiology of fibrosis in both C57BL/6 and FVB, though C57/B6 mice

require higher dose of aristolochic acid (5mg/kg) than FVB mice (4mg/kg) (<https://doi.org/10.1038/ki.2015.51> and doi: 10.1681/ASN.2016121351. Epub 2017 Jul 12)

7) Was proteinuria followed?

Response: We thank the reviewer for raising this important point. We did not measure proteinuria in this study. We previously confirmed increased albuminuria in \square GT-Cre;Tgfr2^{fl/fl} mice compared to Tgfr2^{fl/fl} mice using uninephrectomy/angiotensin II models (UniNX/AngII), a model of renal hyperfiltration and high blood pressure induced CKD leading to remarkable albuminuria (Nlandu-Khodo, Stellor et al. *JASN* vol. 28,12 (2017): 3490-3503. doi:10.1681/ASN.2016121351)

8) At what voltage were the electron micrographs taken? What system, detector?

Response: We apologize for these omissions. Electron microscopy was performed on a Tecnai T12 operating at 100 keV using a side mount AMT CCD camera. This information is added in the online method section "*Mitochondrial Ultrastructure Analysis*"

Figures:

9) Figure 1 and ED (Extended Data) Figure 1: It is not clear what the number of animals was in the experiments that lead to panels 1b and 1c i.e., how variable are the proportions of cell types in the tested groups, and, as the study focuses on male mice, how representative are the findings in general?

Response: We thank the reviewer for raising these important points. We used 6 conditions and one animal per condition. Indeed, the goal of the spatial transcriptomics experiment is to complement our histological findings with spatio-temporal insights on tubulo-interstitial interactions. Therefore, we focused on the GSE151658 dataset in which 8 animals (7 animals for single cell RNAseq and one animal was used for spatial resolution) to specifically identify cell types in our spatial transcriptomics data. After integrating spatial data with single-cell data by applying the 'anchor'-based integration workflow introduced in Seurat v3 ([https://www.cell.com/cell/fulltext/S0092-8674\(19\)30559-8](https://www.cell.com/cell/fulltext/S0092-8674(19)30559-8)) that enables the probabilistic transfer of annotations from a reference to a query set, we had the choice to run Fisher's exact test or a Chi-Square test to indicate in the absence of sample replicates whether a certain cell type is overrepresented in a specific sample. Thus, we applied Fisher's Exact test to indicate

whether a certain cell type is more or less abundant in γ GT-Cre;Tgfb β 2^{fl/fl} sample relative to Tgfb β 2^{fl/fl} as reflected by pValue and oddsRatio. We focused on male mice to follow up our previous study. A gender difference study is already foreseen.

10) Fig 1a: at the resolution provided, the text overlaid onto the UMAP is not legible and, in the case of Myo/St.mixed, seems not to be in the correct location

Response: We apologize for the inconvenience. We removed the text on the UMAP clusters, increased image resolution and kept cluster labelling only on the side legend.

11) Fig 1b: The text overlaid on the spatial feature plot are not legible; the colors for S3 and Endo/Glom are hard to distinguish, especially with the H&E background. It is not clear which time point after aristolochic acid treatment is reflected here.

Response: We thank the reviewer for raising this point. Fig1b depicts the impairment of cluster organization with regard to the renal anatomy layers. We used uninjured and 3 weeks AA injury kidneys. We increased the resolution of images.

12) Fig 1c: colors do not match color scheme in panels a and b

Response: We thank the reviewer for raising this point. We matched colors in panel a, b and c.

13) Fig 1d: the resolution of the H&E and panels below is rather low; the Loupe browser images show horizontal bands with distortions

Response: We thank the reviewer for this comment. The Cloupe browser images in Fig1d provide an overall signal/intensity pattern of injury and fibrosis markers and validate the technique with regard to injury/fibrosis induction. This signal pattern of injury and fibrotic markers is confirmed in Fig.1f and Extended Data Figure 2 for Kim-1 (Havcr1 mRNA and immunofluorescence staining) and Fig.1h for collagens (Sirius red staining).

14) Fig 1e: resolution is marginal

Response: We apologize for the marginal resolution. We increased the resolution of images in the Fig. 1e.

15) ED Fig 1a, b: which genes shown in panel b correspond to which violin plot in panel a? Panels a and b shows 16 clusters. To what areas in the UMAP shown in Fig1a do they map

to i.e., could the authors add a panel in the ED Fig1 that would show the location of the 16 clusters in an UMAP plot?

Response: We thank the reviewer for raising this point. Indeed, Supplementary Fig. 1 shows the prediction score as computed by Seurat's function "TransferData". The label on the y-axis reading "Expression level" represent the "prediction score". The location of the clusters is indicated in the plot below, which comes from the 2021-12-13.Annotate_cellTypes.htm. Please find below the localization of the clusters in the UMAP as included in the Supplementary Fig. 1.

16) Fig 1i, j: in both panels, for the Cre/fl.fl Tgfr2 mice, there are two outliers. Do the data come from the same mice?

Response: We thank the reviewer for raising this point. The outliers come from different mice; however, we added 4 Tgfr2^{fl/fl} and 3 γ GTCre;Tgfr2^{fl/fl} mice in Fig.1j to improve the statistical basis.

17) Figure 2 and ED (Extended Data) Figure 2:

- ED Fig 2b: color scheme makes it hard to identify data points that associate with the term 'Absent'

Response: We thank the reviewer for this comment. We changed the color gray to dark red in the Extended Data Fig. 3 in the revised manuscript.

- Fig 2a: the resolution of the EM images is marginal; the text overlaid on the EM micrographs is not explained in the figure legend; scale-bar dimension for the higher-magnification images is not provided

Response: We thank the reviewer for raising this point and apologize for marginal resolution of images. We increased image resolution, explained the text on the EM micrographs in the appropriate figure legend and added thicker scale bars.

- Fig 2f: how do the data points relate to number of areas per animals per group?

Response: We thank the reviewer for raising this point. For TMRM quantification, 18 different areas from 4 different animals per group were captured and data points relate to TMRM positive area per capture per group. Every dot represents the number of TMRM positive tubules in the field of $900\mu\text{m}^2$.

- Fig 2g: image resolution is marginal

Response: We thank the reviewer for raising this point and apologize for marginal resolution of images. We increased image resolution.

18) Figure 4 and ED (Extended Data) Figure 4:

- ED Fig4: panel d, the purpose of the GFP channel in imaging is not immediately clear from the text or figure legend

Response: We thank the reviewer for raising this point. We assume that the reviewer referred to the Fig. 4d, and we added a description of the mito-QC reporter mouse and the original article in the reference. The mito-QC mice express a pH-sensitive tandem mCherry-GFP tag fused with a fragment (residues 101-152) of the mitochondrial protein Fis1. Under mitophagy, GFP is quenched in the lysosomes and only mCherry is expressed to reflected ongoing mitophagy.

19) Figure 5 and ED (Extended Data) Figure 5:

- ED Fig5: typos in headers; the reviewer does not find the cartoon-like Metacore representation very helpful. In panel a, the 'meter' that shows the expression levels obscures some of the components it associates with; a table format would be easier to follow; however, the depiction of the pathways is appreciated.

Response: We thank the reviewer for this comment. We corrected the typos in headers and we provided additional information in the Supplementary table 4.

20) Figure 6 and ED (Extended Data) Figure 6:

- ED Fig6: the image of the western blot is suggesting that the antibody is detecting two bands for Pgc1- α in T β R11flox/flox; is that typical?

Response: We thank the reviewer for this comment. The datasheet referenced the 113 kDa band as shown in the Extended Data Fig. 8. However, a siRNA targeting Pgc1 α decreased both the 113 kDa and the higher molecular weight band, suggesting that the higher band may be a post-translationally modified Pgc1 α protein. Please find below an example of western blot analysis:

- Fig 6e: typo in header of first panel

Response: We thank the reviewer for this comment. We corrected the typos in Fig. 6e.

21) Figure 7 and ED (Extended Data) Figure 7:

- ED Fig 7: panel a, the resolution of the heat map is marginal, the two groups at 6 weeks are hard to discern by choice of color; I think there are better plotting options to show gene expression along trajectory (?). Panel b, this panel needs more explanation, please add description in text/figure legend; are there data for the 6-week time point? If so, how does it compare with the earlier time point? Panel c needs more information; to a non-initiated it is not clear what LTL is referring to.

Response: We thank the reviewer for raising these points. We changed the colors and improved the resolution of the panel a. For panel b, we added more explanations of cell communication analysis in the figure legend. We used 3 weeks samples in this analysis; this difference is not observed at 6 weeks. For panel c, we described the experiment in the figure legend and defined LTL (Lotus tetragonolobus lectin), which is a marker of proximal tubules.

REVIEWERS' COMMENTS

Reviewer #1 (Remarks to the Author):

This revised paper remains to be important.

The authors did a good job in revising the manuscript and responding the critiques.

I am left with the thought that this manuscript would better shortened and with more clarity.

Reviewer #2 (Remarks to the Author):

For this re-submission, I will recap the summary of this manuscript:

This manuscript by Kayhan et. al seeks to uncover how TGF-B signaling affects PT mitochondria function in CKD. The authors choose to tackle a well-worn topic, but do so with a tour de force approach, comprehensively defining the metabolic perturbations, transcriptomic alterations, and injury phenotypes of TGFBR2 knockout in an aristolochic acid animal model. The authors perform additional validation in a proximal tubule cell culture model and draw upon existing clinical datasets to provide further corroboration of their results. This work builds upon their prior investigations in a 2017 JASN publication using the same murine model. The authors conduct a bevy of experiments and analyses: 1) histology and BUN, 2) immunofluorescence of immune cell infiltration and Pgc1a, 3) spatial transcriptomics, 4) use

publicly available sc or sn RNAseq of murine and human kidneys including cell chat analyses 5) electron microscopy of mitochondria, 6) FACS for immune cell infiltration, 7) intravital

imaging for dextran uptake, 8) Seahorse cell culture, 9) RNAseq of a PT cell culture model with TGFBR2 knock out, 10) metabolic bioluminescence assays, 11) DCFDA cell assay, 12) RT-PCR and immunoblotting of key genes and proteins. The strengths of this investigation include strong orthogonal validation of their murine model with a vast array of techniques. The

effect of TGFBR2 KO in a CKD model is comprehensively investigated, mixing recent technologies like spatial transcriptomics with classic ones. The potential of this manuscript has been nearly maximized.

The authors were mostly responsive and addressed about 40 critiques.

A major critique remains. The examination of mitochondrial dysfunction associated with TGF-beta in kidney disease has been well-studied and many of the techniques utilized in this manuscript have been previously published. The main advancement of science in this study is the inclusion of scRNAseq and spatial transcriptomics to study this pathway. However, the scRNAseq is not model specific and the spatial transcriptomics data are not analyzed at a state-of-the-art level that Nature Communications readers might expect. The authors show broad up-regulation of injury markers without niche analyses or deconvolution. I encourage the authors to examine this publication by Kuppe et. al (PMID: 35948637) to see how the Visium technology is leveraged to answer colocalization and niche questions. It would still be of significant interest to understand how immune cell infiltrate (like Th1 cells) localize with the various epithelial cell types (like PT S3).

As it stands, there may be insufficient justification for the use of Visium in this manuscript. The injury markers and pathways could have been appreciated in bulk sequencing. A novel injury spot or cell cluster is not identified. Further, there is no localization of injury. It does not seem quite enough to complete the wet lab portion of Visium, without the relevant analyses included as well.

Reviewer #3 (Remarks to the Author):

The authors comprehensively addressed the reviewer's comments. The reviewer does not have any further concerns and recommends the manuscript for publication.

In response to the reviewers' comments, the following changes have been made:

Reviewer #1

This revised paper remains to be important. The authors did a good job in revising the manuscript and responding the critiques. I am left with the thought that this manuscript would better shortened and with more clarity.

Response: We appreciate the reviewer's comment. In order to comply with Nature Communications broad audience, we avoided usage of short technical words keeping the manuscript more detailed than it would be in a specialized journal.

Reviewer #2

For this re-submission, I will recap the summary of this manuscript:

This manuscript by Kayhan et. al seeks to uncover how TGF-B signaling affects PT mitochondria function in CKD. The authors choose to tackle a well-worn topic, but do so with a tour de force approach, comprehensively defining the metabolic perturbations, transcriptomic alterations, and injury phenotypes of TGFBR2 knockout in an aristolochic acid animal model. The authors perform additional validation in a proximal tubule cell culture model and draw upon existing clinical datasets to provide further corroboration of their results. This work builds upon their prior investigations in a 2017 JASN publication using the same murine model. The authors conduct a bevy of experiments and analyses: 1) histology and BUN, 2) immunofluorescence of immune cell infiltration and Pgc1a, 3) spatial transcriptomics, 4) use publicly available sc or sn RNAseq of murine and human kidneys including cell chat analyses 5) electron microscopy of mitochondria, 6) FACS for immune cell infiltration, 7) intravital imaging for dextran uptake, 8) Seahorse cell culture, 9) RNAseq of a PT cell culture model with TGFBR2 knock out, 10) metabolic bioluminescence assays, 11) DCFDA cell assay, 12) RT-PCR and immunoblotting of key genes and proteins. The strengths of this investigation include strong orthogonal validation of their murine model with a vast array of techniques. The effect of TGFBR2 KO in a CKD model is comprehensively investigated, mixing recent technologies like spatial transcriptomics with classic ones. The potential of this manuscript has been nearly maximized.

The authors were mostly responsive and addressed about 40 critiques.

Response: We thank the reviewer for recognizing the questions addressed.

A major critique remains. The examination of mitochondrial dysfunction associated with TGF-beta in kidney disease has been well-studied and many of the techniques utilized in this manuscript have been previously published. The main advancement of science in this study is the inclusion of scRNAseq and spatial transcriptomics to study this pathway. However, the scRNAseq is not model specific and the spatial transcriptomics data are not analyzed at a state-of-the-art level that Nature Communications readers might expect. The authors show broad up-regulation of injury markers without niche analyses or deconvolution. I encourage the authors to examine this publication by Kuppe et. al (PMID: 35948637) to see how the Visium technology is leveraged to answer colocalization and niche questions. It would still be of significant interest to understand how immune cell infiltrate (like Th1 cells) localize with the various epithelial cell types (like PT S3).

As it stands, there may be insufficient justification for the use of Visium in this manuscript. The injury markers and pathways could have been appreciated in bulk sequencing. A novel injury spot or cell cluster is not identified. Further, there is no localization of injury. It does not seem quite enough to complete the wet lab portion of Visium, without the relevant analyses included as well.

Response: We thank the reviewer for this important comment. Mitochondrial dysfunction (mostly increased ROS production) associated with TGF- β has been reported in vitro; however the mechanisms whereby TGF- β affects mitochondrial function remained speculative in previous studies. Our study pointed out the dose-dependence effect of TGF- β on mitochondrial function such as lower doses promote oxidative phosphorylation whereas higher doses impair it; supporting the fact that intrinsic TGF- β signaling is beneficial in proximal tubule response to chronic renal injury. Most of in vitro data demonstrating the harmful effect of TGF- β used extremely high doses of TGF- β (10-30 ng/ml) leading to limited interpretation of TGF- β effect in these studies. In CKD, studies established association between high systemic TGF- β , renal fibrosis and mitochondrial dysfunction. In this study, we established how intrinsic TGF- β signaling modulates mitochondrial integrity and function under acute to chronic renal injury in the most metabolic and vulnerable renal segment (the proximal tubule). Regarding the Transcriptomics, we agreed with the reviewer that not having generated our scRNAseq data constituted a limitation in spatial transcriptomics analysis including cell deconvolution. We therefore clearly stated this limitation in the discussion. Using integration

over deconvolution methods in our spatial transcriptomics, the output for each spot is a probabilistic classification for each of the scRNA-seq derived cell types, meaning that the reported cell type represents the dominant cell type at each spot. The goal of the spatial transcriptomics was not to identify injury site which is clearly known in the field (that we can also determine by marker expression per annotated clusters), but to determine cell type/annotated cluster interaction using the CellchatDB. This procedure allowed us to point out enhanced interaction between the S3T2 cells and Macrophage and dendritic cell population in line with our FACS and IHC/immunofluorescence findings. Moreover, a separate spatial transcriptomics manuscript is already foreseen as a follow up of this study to uncover molecular contributors involved in the S3T2 and immune cell interaction.

Reviewer #3

The authors comprehensively addressed the reviewer's comments. The reviewer does not have any further concerns and recommends the manuscript for publication.

Response: We thank the reviewer for the decision.